# The coilin N-terminus mediates multivalent interactions between coilin and Nopp140 to form and maintain Cajal bodies

Edward Courchaine[1,2], Sara Gelles-Watnick [1,2], Martin Machyna[1,2], Korinna Straube[1], Sarah Sauyet [1], Jade Enright [1] & Karla M. Neugebauer [1] ✉

Cajal bodies (CBs) are ubiquitous nuclear membraneless organelles (MLOs) that concentrate and promote efficient biogenesis of snRNA-protein complexes involved in splicing (snRNPs). Depletion of the CB scaffolding protein coilin disperses snRNPs, making CBs a model system for studying the structure and function of MLOs. Although it is assumed that CBs form through condensation, the biomolecular interactions responsible remain elusive. Here, we discover the unexpected capacity of coilin's N-terminal domain (NTD) to form extensive fibrils in the cytoplasm and discrete nuclear puncta in vivo. Single amino acid mutational analysis reveals distinct molecular interactions between coilin NTD proteins to form fibrils and additional NTD interactions with the nuclear Nopp140 protein to form puncta. We provide evidence that Nopp140 has condensation capacity and is required for CB assembly. From these observations, we propose a model in which coilin NTD–NTD mediated assemblies make multivalent contacts with Nopp140 to achieve biomolecular condensation in the nucleus.

Cajal bodies (CBs) are membraneless organelles (MLOs) present in the nuclei of most vertebrate, insect, and plant cells with a catalog of molecular components[1,2]. Discovered by Ramon y Cajal in 1903, CBs are an important model for understanding the assembly, maintenance, and function of MLOs due to extensive knowledge about their composition and function. CBs are the sites of small non-coding ribonucleoprotein particle (RNP) assembly and recycling. Nucleolar snoRNAs and spliceosomal snRNAs traffic through CBs for snRNA modification by scaRNPs and maturation to functional components of the splicing and ribosomal processing machinery, namely snRNPs and snoRNPs[3–9]. CBs are also nucleated at specific sets of genes, such as those encoding snRNAs, thereby clustering whole chromosomal regions within the three-dimensional space of the nucleus[3,10–12]. Like other MLOs, CBs are dynamic and contain numerous molecular constituents that freely diffuse in and out of CBs with residence times of less than 1 minute[5,13,14]. CBs disassemble before mitosis and assemble thereafter, are absent in G0, and are modified to respond to cellular and organismal stress[15].

Unlike many other MLOs, at least one important function of CBs themselves is known. The ability to deplete or genetically delete the canonical scaffolding factor of CBs, coilin, provides a handle for identifying deficits upon CB loss in plants and animals in vivo[16–20]. Some activities, like snRNA modification by scaRNPs, occur in the absence of coilin due to the persistence of residual bodies[8,17]. Although phenotypes differ among species, coilin is essential for vertebrate embryonic survival and for mammalian fertility[16,17,21–23]. Loss of viability due to inefficient snRNP assembly and splicing supports evidence that the rate of snRNP assembly is faster in CBs, allowing the biogenesis of components of the splicing machinery to keep pace with zygotic gene expression during very short cell cycles[16,24,25]. Moreover, CB composition and morphology are significantly altered in human disease, such as Spinal Muscular Atrophy (SMA) and Down Syndrome[26–28].

Despite this wealth of information and their obvious physiological importance, the field lacks a mechanistic understanding of how CBs assemble. Because of the absolute requirement for coilin, the

---

[1]Department of Molecular Biophysics and Biochemistry, Yale University, New Haven, CT, USA. [2]These authors contributed equally: Edward Courchaine, Sara Gelles-Watnick, Martin Machyna. ✉e-mail: karla.neugebauer@yale.edu

answer must lie with its structure and function. Coilin contains three regions that are separable by the degree of evolutionary conservation[2]. The N- and C-terminal domains (NTD and CTD, respectively) are the most highly conserved; they are separated by a region of low complexity that is poorly conserved and likely intrinsically disordered. This middle domain contains a bipartite nuclear localization signal (NLS) and an RG box modified by dimethylarginine and bound by the survival of motor neuron protein (SMN) tudor domain[13,29–32]. The coilin CTD harbors its own tudor domain, which interestingly lacks the aromatic amino acids that would enable it to bind dimethylarginine[33]. Instead, the coilin CTD binds to the core Sm ring of spliceosomal snRNPs that are defining components of CBs[30,34,35]. The coilin CTD likely binds several other proteins and determines the number of CBs per cell[2,36,37].

The coilin NTD alone can localize to CBs formed by full-length coilin, and coilin constructs lacking the NTD cannot[37]. The same study showed that the NTD scored positively in a yeast two-hybrid screen with full length coilin, and the NTD was also required for the pull-down of endogenous coilin from cell extracts. Since then, the NTD has been referred to as a self-association domain. Indeed, coilin does self-associate both in nucleoplasm and in CBs, as detected by fluorescence resonance energy transfer (FRET)[4]. However, the region on full length coilin that interacts with the NTD has not been identified, and the role of the NTD in CB assembly is unknown. Thus, it has never been clear if coilin NTD binds to the NTDs of other coilin molecules, if it can form higher order multimers, or how any of its potential activities support de novo CB assembly. Therefore, the present study strives to answer these mechanistic questions and to

determine if an NTD–NTD interaction is both necessary and sufficient to drive CB formation.

Here, we dissect the molecular functions of the coilin NTD in vivo, expressing NTD constructs in cells lacking endogenous coilin (*Coil*−/− MEFs[17]) as a model for steps in de novo CB assembly. Nuclear and cytoplasmic expression of the coilin NTD were used to sample the ability of the coilin NTD to self-associate in different cellular environments, enabling our discovery that the NTD is capable of forming fibrils and identifying Nopp140 as a critical nuclear interaction partner of coilin in forming CBs. Expression of NTD constructs harboring alanine point mutations combined with indirect FRET in vivo enabled us to probe the specificity of interactions and the morphologies of the structures formed. Finally, the use of cells containing endogenous coilin, and therefore pre-existing CBs, enabled us to infer assembly and maintenance principles, by providing a test for whether mutants can interact with endogenous coilin and if their multivalency supports dominant negative phenotypes. The use of protein structure prediction to visualize the positions of the amino acids critical for fibril formation and Nopp140 interaction by the coilin NTD allowed us to propose a working model for how the interplay between coilin–coilin and coilin–Nopp140 interactions contribute to CB assembly.

## Results

### Coilin NTD assembly properties distinct from dimerization

To investigate the hypothesis that the coilin NTD promotes CB assembly through dimerization activity, we asked if the NTD could be replaced by a heterologous protein dimerization domain. Thus, we cloned a set of expression constructs (Fig. 1a): full-length coilin, coilin

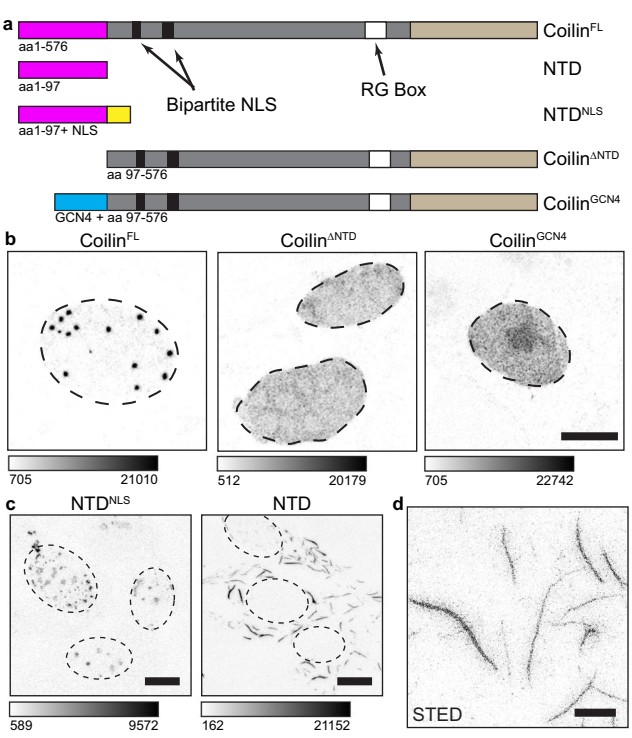

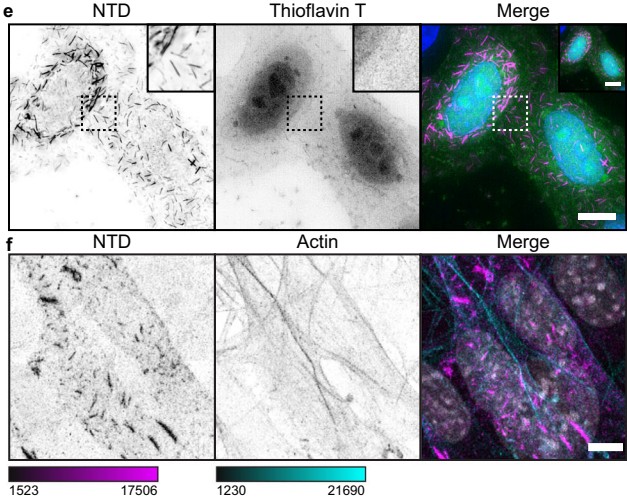

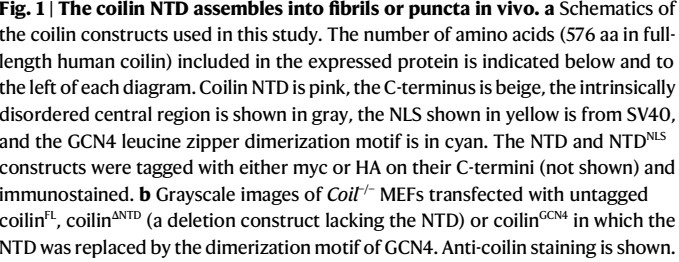

**Fig. 1 | The coilin NTD assembles into fibrils or puncta in vivo. a** Schematics of the coilin constructs used in this study. The number of amino acids (576 aa in full-length human coilin) included in the expressed protein is indicated below and to the left of each diagram. Coilin NTD is pink, the C-terminus is beige, the intrinsically disordered central region is shown in gray, the NLS shown in yellow is from SV40, and the GCN4 leucine zipper dimerization motif is in cyan. The NTD and NTD^NLS constructs were tagged with either myc or HA on their C-termini (not shown) and immunostained. **b** Grayscale images of *Coil*−/− MEFs transfected with untagged coilin^FL, coilin^ΔNTD (a deletion construct lacking the NTD) or coilin^GCN4 in which the NTD was replaced by the dimerization motif of GCN4. Anti-coilin staining is shown.

**c** Grayscale images NTD and NTD^NLS transiently expressed in *Coil*−/− MEFs. **d** STED image of coilin NTD transfected *Coil*−/− MEFs. Scalebar = 2 μm. **e** Cytoplasmic NTD-myc fibrils (magenta) in HeLa cells do not stain with Thioflavin T (green). DNA visualized with Hoechst (blue). **f** *Coil*−/− MEFs transfected with NTD-myc (magenta) and counterstained with α-actin (cyan) to compare actin filaments with cytoplasmic coilin NTD fibrils. DNA visualized with Hoechst (gray). This experiment was repeated independently two times with similar results. All imaged with Leica SP8 laser scanning confocal microscope. Color bars are in analog-digital units to indicate differences in expression. Dotted lines in **b**, **c** indicate the nuclear limits. Scalebars (all except **d**) = 10 μm; insets, 2 μm.

without the N-terminal domain, and coilin with a dimerization domain (the leucine zipper motif of GCN4) instead of the NTD[38]. Each construct was transfected into mouse embryonic fibroblasts derived from coilin gene knock-out mice (*Coil*[-/-] MEFs), which thereby lack CBs[17]. Each protein was expressed to similar levels (Supplementary Fig. 1a). FRET assays detected significant interactions between full-length coilin molecules (coilin[FL]–coilin[FL]), as shown previously[4], and between those dimerized by the GCN4 leucine zipper (coilin[GCN4]-coilin[GCN4]) confirming that the GCN4 leucine zipper induces coilin dimerization (Supplementary Fig. 1b). As expected, coilin[FL] induced the formation of CBs in *Coil*[-/-] MEF nuclei (Fig. 1b). Expression of coilin[ΔNTD] generated diffuse nuclear signal with no nuclear puncta, indicating the requirement of the coilin NTD for de novo CB assembly, not just localization as shown previously[37].

If NTD mediated coilin dimer- or multimerization is important for CB assembly, we would expect its deletion to affect coilin–coilin FRET signals in cells. Importantly, coilin–coilin FRET was abolished upon NTD deletion, indicating that the NTD is required for coilin intermolecular association (Supplementary Fig. 1b). To test if dimerization alone is sufficient, the ability of coilin[GCN4] to assemble CBs was tested. Strikingly, coilin[GCN4] dimers recapitulated the diffuse stain of coilin[ΔNTD], rather than that of coilin[FL]. The diffuse phenotypes of coilin[ΔNTD] and coilin[GCN4] occurred at a wide range of cellular expression levels (Supplementary Fig. 1c). The observed diffuse localization of both constructs differs from the typical phenotype of 2–4 CBs that are 0.5–1.0 μm in diameter. We conclude that coilin dimerization by the N-terminus is not sufficient for CB formation. Rather, coilin NTD may facilitate CB formation through other NTD interactions in addition to dimerization.

To investigate the independent function(s) of the coilin NTD isolated from the rest of the coilin molecule, which has numerous interaction partners[2], we expressed the coilin NTD alone in the *Coil*[-/-] MEF cytoplasm or nucleus utilizing a viral SV40 nuclear localization signal (Fig. 1a). In the nucleus, NTD[NLS] formed bright puncta and also localized diffusely to nucleoli (Fig. 1c and Supplementary Fig. 1d). Unexpectedly, the NTD construct lacking the NLS formed abundant, extended structures up to 10 micrometers long in the cytoplasm (Fig. 1c). Despite this length, the structures appear to be as little as 100 nm in diameter by sub-diffraction imaging (Fig. 1d). Based on these dimensions and their similarity to collagen and other fibrils, we have chosen to refer to these structures as fibrils throughout the manuscript. Aggregation-sensitive dyes Thioflavin T (recognizes amyloid), Congo Red (recognizes amyloid), and ProteoStat (recognizes aggregates) failed to stain cytoplasmic NTD fibrils (Fig. 1e and Supplementary Fig. 1e). Therefore, the fibrils do not appear to be amyloid in nature. Additionally, the fibrils do not form along actin filaments, which appear quite distinct and non-overlapping (Fig. 1f). We conclude that coilin NTD forms fibrils when expressed in the cytoplasm, which may be homo-multimers or may contain other components.

Interestingly, the nuclear-expressed NTD[NLS] protein localized to round puncta that appeared to be similar in size to CBs formed by transfection of coilin[FL] (compare Fig. 1b, c). Coilin[FL] colocalized with typical snRNP components of CBs (Supplementary Fig. 1f), as expected[4,17]. However, the NTD[NLS] puncta did not contain canonical CB components such as SART3, SMN, or SmB″ (Supplementary Fig. 1g); this was expected, because the coilin C-terminus is required for snRNP and SMN recruitment[35,36]. Thus, the large, nuclear NTD[NLS] puncta are not bona fide CBs. Nevertheless, a notable exception was Nopp140 (Supplementary Fig. 1g), which did colocalize with NTD[NLS]. Nopp140 is a nucleolar and CB protein previously shown to interact with coilin NTD[39,40] that has been linked with the accumulation of snoRNP components and scaRNPs in CBs and residual bodies[6,17,41]. The snoRNP component, fibrillarin, was broadly nucleoplasmic and not well-localized to the NTD[NLS] puncta, suggesting the NTD[NLS] puncta are not limited to possible residual bodies observed previously in these cells[17].

Taken together, we conclude that the nuclear environment supports the formation of NTD puncta, as opposed to fibrils, and contains at least one other NTD interaction partner, Nopp140.

The observed colocalization of Nopp140 with nuclear NTD[NLS] in puncta raised the question of whether the coilin NTD–Nopp140 interactions contribute to CB assembly, which is a controversial point in the literature. Partial knockdown of Nopp140 by stable CRISPR/Cas9 editing did not disturb CB formation[6,41], while two other studies showed that loss of Nopp140 is correlated with CB disassembly[40,42]. To determine whether Nopp140 is required for CB assembly and maintenance in our system, we depleted Nopp140 and coilin proteins from HeLa cells using siRNAs (Fig. 2a; note that the NOLC1 gene encodes Nopp140 protein). Both Nopp140 and coilin were poorly detected by immunofluorescence after knockdown, confirming efficient transfection of the siRNAs in the cell population (Fig. 2b, c and Supplementary Fig. 2a). CB numbers were broadly reduced after Nopp140 depletion based on coilin and tri-methylguanosine (TMG, a marker for snRNAs) immunostaining (Fig. 2b–d and Supplementary Fig. 2b). Gems marked by SMN appeared relatively unaffected (Fig. 2b and Supplementary Fig. 2a).

To quantify the effects of Nopp140 depletion on nuclear MLOs, we used an automated pipeline (see "Methods") for counting the number of CBs per nucleus (Fig. 2e); in addition, we counted CBs and gems manually (Supplementary Fig. 2c, d). The automated approach detects ~2 CBs/nucleus in HeLa cells transfected with *siNT* (median 2, mean, 1.6, std 1.2); 0–1 CBs/nucleus with *siCoil* (median 0, mean 1.0, std 1.4); and 0–0.8 CBs/nucleus with *siNolc1* (median 0, mean 0.8, std 1.2). Importantly, both automated and manual counts indicate that the reduction in CB numbers upon depletion of Nopp140 is statistically significant. Gems can be closely associated with CBs[32,43] and do not depend on the presence of coilin for assembly and maintenance[17]; in agreement, there are only marginal changes in the number of gems per nucleus upon coilin or Nopp140 depletion (Supplementary Fig. 2d). Taken together, these data indicate that Nopp140 is necessary for the assembly and/or maintenance of CBs containing coilin and snRNPs.

## Nopp140 is required for coilin NTD condensation into puncta

Although CBs are often included in lists of biomolecular condensates[44–46], only scant experimental evidence of CB condensation exists. One possibility is that CB condensation occurs via multivalent interactions of the intrinsically disordered region of coilin (Fig. 3a). Alternatively, Nopp140 (which is predicted to be almost entirely intrinsically disordered) might engage in multivalent interactions with coilin and confer condensation properties on the complex. We took advantage of the optodroplet assay[32,47], which relies on light activation of the Cry2 dimerization domain to trigger a local increase in the Cry2 fusion protein concentration (Supplementary Fig. 3a–c). Coilin[FL] makes CBs with and without optical activation, as expected (Supplementary Fig. 3c). In contrast, coilin[ΔNTD]-Cry2 fails to form condensates in the absence or presence of light (Fig. 3b). Therefore, the IDR and C-terminal regions of coilin do not have propensity for biomolecular condensation. These data suggest that the IDR of coilin is not responsible for Cajal body condensation. Importantly, Nopp140[IDR]-Cry2 readily formed optodroplets (Fig. 3c), indicating that the Nopp140 IDR has the capacity for condensation. To test whether Nopp140 is responsible for the condensation of the NTD[NLS] puncta, Nopp140 was depleted from HeLa cells transfected with coilin NTD[NLS]. As predicted, NTD[NLS] no longer formed puncta (Fig. 3d). Instead, we observed fibrils localized to the nucleoli. This suggests that Nopp140 interaction is responsible for remodeling coilin NTD–NTD assemblies in the nucleus to puncta.

## Residues involved in coilin, Nopp140 interactions identified

To characterize coilin–Nopp140 interactions and gain mechanistic insight into how the coilin NTD can attain morphologies

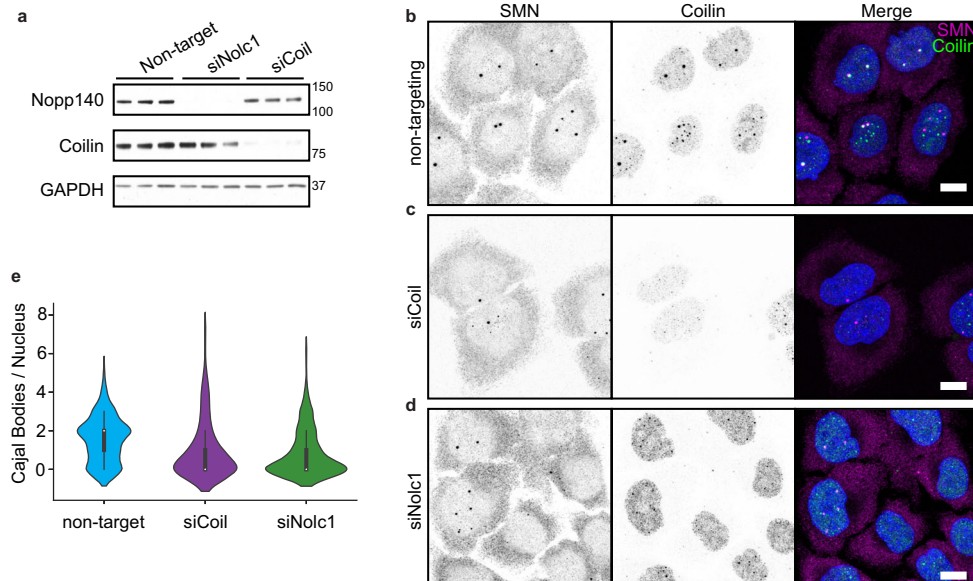

**Fig. 2 | Nopp140 is essential for Cajal body assembly. a** Western blots of total lysate from three biological replicates of HeLa cells depleted of Nopp140 or coilin using siRNA transfection targeting *Nolc1* and *coil* mRNAs, respectively. Molecular weight markers indicated to the right of the blots. Representative images for cells undergoing transfection with a non-targeting siRNA (**b**), si*Coil* oligo pool (**c**), or si*Nolc1* oligo pool (**d**). Scale bars = 10 μm. SMN and Coilin signal imaged under identical conditions and displayed at the same levels. **e** Violin plots of Cajal body count per nucleus based on automated quantification of coilin immunostaining. *N* = 140 (non-targeting siRNA); 103 (si*Coil*); 142 (si*Nolc1*). Violin plot area is normalized between samples, white dots represent the median, box denotes the interquartile range, and the whiskers note 1.5 the interquartile range. Source data are provided as a Source Data file.

characteristic of fibrils and puncta, we performed an alanine mutagenesis screen of 29 highly conserved amino acids present in the NTD (red shading in Fig. 4a and Supplementary Fig. 4). Each full length coilin variant containing a single alanine point mutation was transiently expressed in *Coil*[-/-] MEFs and immunostained for total coilin (Supplementary Fig. 5a). Many of the conserved amino acids occur within predicted beta sheet and alpha helical motifs, together identified as a ubiquitin-like fold according to the RaptorX algorithm (Supplementary Fig. 4). Figure 4b shows that each of three single amino acid mutants R8A, R36A, and D79A led to noticeable morphological differences in the MLOs formed by coilin[FL]. As before, wild-type (WT) coilin[FL] formed large, discrete CBs with little observable fluorescence in the surrounding nucleoplasm (Figs. 1b and 4b). By comparison, R8A and D79A yielded small puncta and hazy nucleoplasmic stain; R36A was chosen for contrast, because it had a subtler phenotype. Other alanine point mutants had a wide range of phenotypes (Supplementary Fig. 5a).

To measure changes in coilin–coilin and/or coilin–Nopp140 interactions associated with any of the single amino acid mutations in the NTD, we used our acceptor photobleaching FRET assay (see Supplementary Fig. 1b). Wild-type coilin–coilin FRET yielded ~20% apparent FRET, while Nopp140–coilin FRET yielded ~35% (Fig. 4c, d). Interestingly, the R8A mutation nearly abolished coilin–coilin FRET and D79A reduced it. This loss of FRET indicates regions on the coilin NTD sequence and predicted structure that could mediate coilin multimerization through NTD–NTD interactions. Single amino acid mutations also perturb Nopp140–coilin interactions, because R8A and R36A abolished Nopp140–coilin FRET, while D79A reduced FRET.

To validate the interactions detected by FRET, co-immunoprecipitations were conducted from cells expressing GFP-tagged coilin. Wild-type coilin–GFP efficiently pulled down endogenous coilin and Nopp140 from a cellular extract (Fig. 4e). To control for the possibility that expression of these variants could alter the soluble fraction of endogenous coilin or Nopp140, western blotting of soluble (input) and insoluble fractions was performed (Supplementary Fig. 5b). In agreement with the FRET measurements, R8A and D79A

mutations pulled down reduced amounts of endogenous coilin. In contrast, coilin R36A-GFP pulled down endogenous coilin amounts similar to WT coilin-GFP. Yet, coilin-GFP constructs harboring any of the three mutations failed to pull down Nopp140. We used the RaptorX[48] algorithm to predict the tertiary structure of coilin NTD and inspected the positions of amino acids R8, R36, and D79 (Fig. 4f). The predicted structure suggests that R36 is accessible to Nopp140 on one face of the NTD, while R8 and D79 are accessible to other coilin molecules on another face. The FRET, co-IP, and structure prediction together suggest that R8 and D79 mediate coilin–coilin interactions while R36 mediates coilin–Nopp140 interactions. We speculate that the Nopp140–coilin interaction may depend on coilin–coilin multimerization, because the R8A and D79A mutations affect both coilin–coilin and Nopp140–coilin interactions.

### Puncta depend on coilin NTD–NTD and NTD–Nopp140 interactions

We next probed if NTD mutants with reduced coilin–coilin and coilin–Nopp140 interaction affected the tendency towards a fibrillar versus punctate morphology. Specifically, mutations that disrupt coilin NTD–Nopp140 interactions might be expected to produce nuclear coilin NTD fibrils similar to NTD fibrils in the cytoplasm (see Fig. 1c, d) or Nopp140 depletion (see Fig. 3d). Since NTD–NTD FRET efficiencies were reduced upon R8A and D79A mutation while R36A had no effect (Supplementary Fig. 6a), we postulated that R8 and D79 mediate coilin self-association while R36 mediates coilin–Nopp140 interactions. To assess fibril formation, wild-type and mutant NTD or NTD[NLS] were transfected into *Coil*[-/-] MEFs cells, as done previously (see Fig. 1a). All constructs were transfected and expressed at similar levels except R8A, where many fewer cells were transfected successfully (Supplementary Fig. 6b–e). Wild-type NTD was localized to the cytoplasm and formed fibrils, as expected (Fig. 5a). R8A and D79A mutations abolished fibril formation, in agreement with our model. Interestingly, the NTD containing the R36A mutation formed WT-like cytoplasmic fibrils, consistent with FRET measurements indicating that R36 does not mediate coilin self-association.

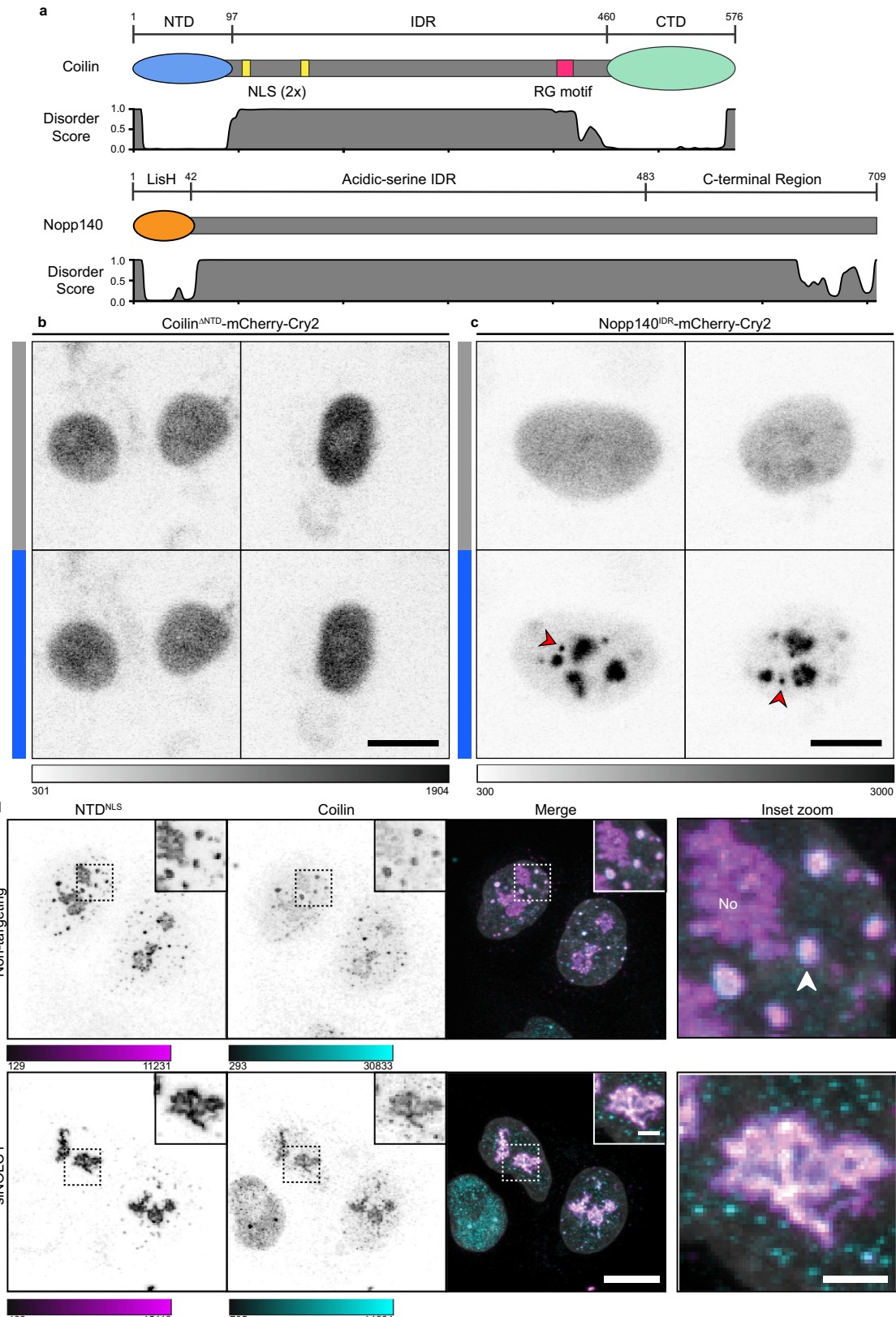

**Fig. 3 | Nopp140 forms inducible condensates and is required for nuclear NTD puncta, suggesting a mechanism for Cajal body condensation. a** Domain architectures of coilin and Nopp140 and predicted disorder score from RaptorX algorithm. **b** NIH-3T3 cells expressing Coilin$^{\Delta NTD}$ mCherry-Cry2 construct. Blue bars indicate 3-minute light activation of the Cry2 domain. **c** NIH-3T3 cells expressing Nopp140$^{IDR}$ mCherry-Cry2 construct. Red arrowheads indicate non-nucleolar bodies formed by blue light induction. **d** HeLa cells transfected with siRNA, non-targeting (top) and NOLC1 (bottom). Cells co-transfected with coilin NTD$^{NLS}$ (magenta) and counterstained for coilin (cyan). DNA visualized with Hoechst (gray). "No" denotes the nucleolus, white arrow denotes a representative Cajal body. Imaged with Leica SP8 laser scanning confocal microscope. Grayscale or color bars are given in analog-digital units. Scale bars = 10 μm; insets, 2 μm; Inset zoom, 2 μm.

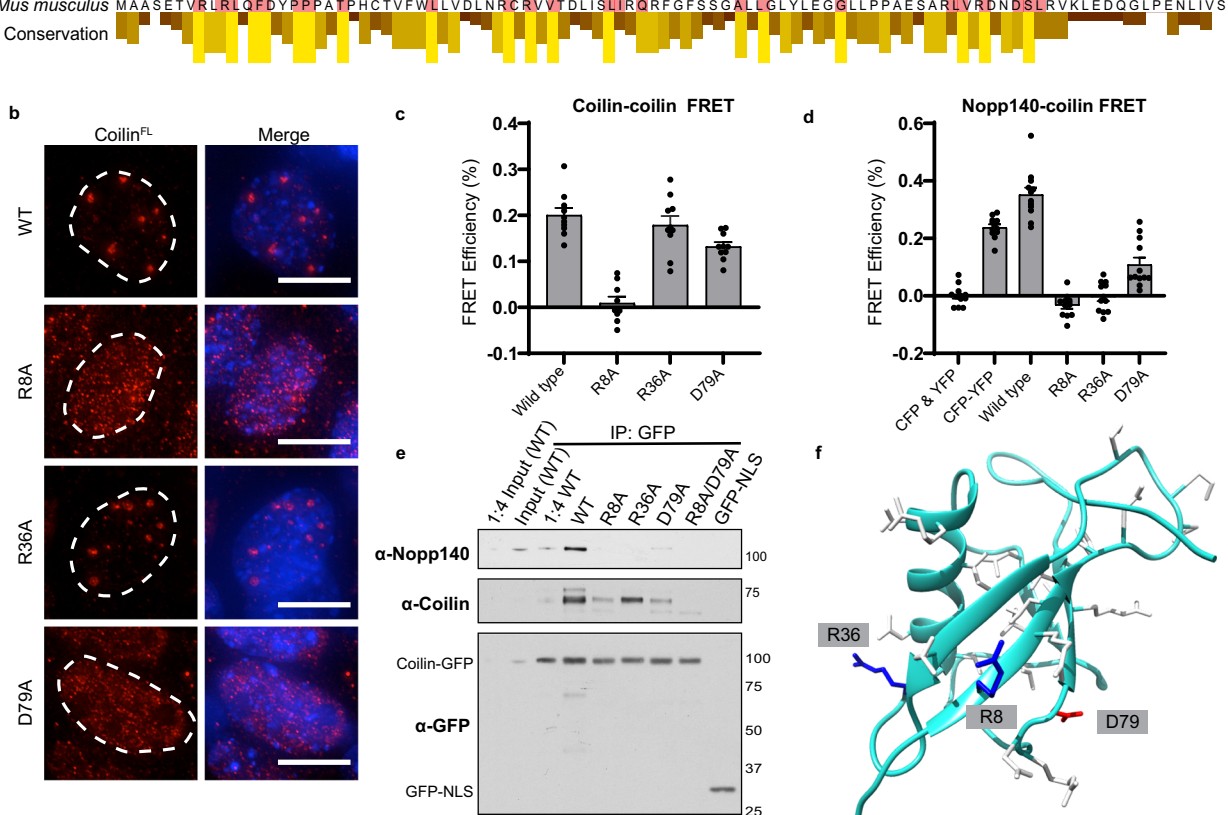

**Fig. 4 | Point mutations in the coilin NTD have distinct effects on self-association and Nopp140 interactions. a** Sequence alignment between mouse and human coilin NTDs with conservation scores from a Clustal Omega multiple alignment of vertebrate coilin sequences. Residues highlighted in red with high conservation scores were selected for alanine mutagenesis. **b** Wild-type or mutated coilin^FL was transfected into *Coil*^−/− MEFs and visually examined for CB formation (coilin is red, DNA is blue). Scale bars = 10 μm. Images acquired with DeltaVision. **c**, **d** In vivo acceptor photobleaching FRET measurements yield apparent FRET efficiencies calculated for constructs co-expressed in HeLa cells. FRET measurements between (**c**) coilin-CFP as the donor and coilin-YFP as the acceptor and **d** between coilin-CFP (donor) and Nopp140-YFP (acceptor). The indicated alanine mutations were present in all coilin molecules tested. Data is represented as mean values ± S.E.M. $N = 10$ (except R8A, where $n = 9$) (**c**) or 12 (**d**) cells measured. Black dots indicate individual data points. **e** Co-immunoprecipitation performed from HeLa cell total lysate after transfection with wild-type and mutant coilin-GFP constructs. Coilin-GFP acts as the bait for either endogenous coilin or Nopp140 as prey and detected by Western blotting as indicated in bold. Molecular weight markers are indicated to the right of the blots. **f** RaptorX-predicted structure of human coilin NTD, a ubiquitin-like fold. Amino acids implicated in coilin–coilin and coilin–Nopp140 interaction are represented as sticks. Source data are provided as a Source Data file.

In the nucleus, wild-type NTD^NLS and Nopp140 formed puncta and some NTD^NLS was localized to nucleoli (Fig. 5a, b; see Supplementary Fig. 1d). The R8A and D79A mutations disrupted all puncta, which we attribute to the loss of coilin self-association observed by FRET. Reinforcing these data, we found that the NTD^NLS has a lower saturation concentration than NTD^NLS R8A or D79A (Supplementary Fig. 6d). Remarkably, the R36A mutation, which reduces Nopp140–coilin but not coilin–coilin interactions (see Fig. 4c), caused NTD^NLS fibrils to form in the nucleus (Fig. 5a) and to decrease saturation concentration below wild-type NTD^NLS! These fibrils did not colocalize with Nopp140 outside of the nucleolus (Fig. 5b), in agreement with our prediction that the R36A mutation disrupts Nopp140–coilin interaction. Thus, the nuclear localization of Nopp140 (see Supplementary Fig. 2a) explains why the wild-type NTD forms fibrils only in the cytoplasm. Moreover, the observation that R8 and D79 are essential for the formation of cytoplasmic fibrils, as well as nuclear puncta, agrees with our proposal that these amino acids mediate coilin NTD–NTD interactions.

### Endogenous coilin has the capacity to form fibrils

The dramatic effects of the single amino acid mutations on the formation of coilin NTD assemblies in *Coil*^−/− MEF cells–in the absence of endogenous coilin–led us to wonder if we could use these WT and mutant NTD constructs to probe interactions with endogenous coilin in cells with pre-existing CBs (Fig. 6a–d). For example, if expression of R36A mutant of the NTD constructs were to form fibrils in HeLa cells, we would expect endogenous coilin to be recruited to these fibrils though the R8 and D79 coilin–coilin interface we provided evidence for above. Remarkably, both R36A and endogenous coilin colocalized into hybrid nuclear fibrils when NTD^NLS R36A was transfected into HeLa cells, as expected (Fig. 6d). We observed a similar phenotype in the cytoplasm when NTD R36A is transfected (Fig. 6c). Thus, the NTD can form multimers with endogenous coilin, demonstrating the ability of the WT coilin molecule to form fibrils when the interactions of the multimer with Nopp140 are reduced.

Transfection of NTD and NTD^NLS mutants R8A and D79A abolished endogenous CBs and caused all coilin to disperse (Fig. 6a–d). In contrast, the wild-type NTD forms oligomers in the cytoplasm of HeLa cells similar to those we observed in mouse cells, while the NTD^NLS protein readily enters and concentrates in CBs and nucleoli (Fig. 6b, d). Transfection of wild-type NTD and NTD^NLS constructs do not significantly alter the number of Cajal bodies per nucleus (Fig. 6a, c). The dominant negative disassembly of all CBs by R8A or D79A suggests that each mutant is able to interact with endogenous coilin but unable

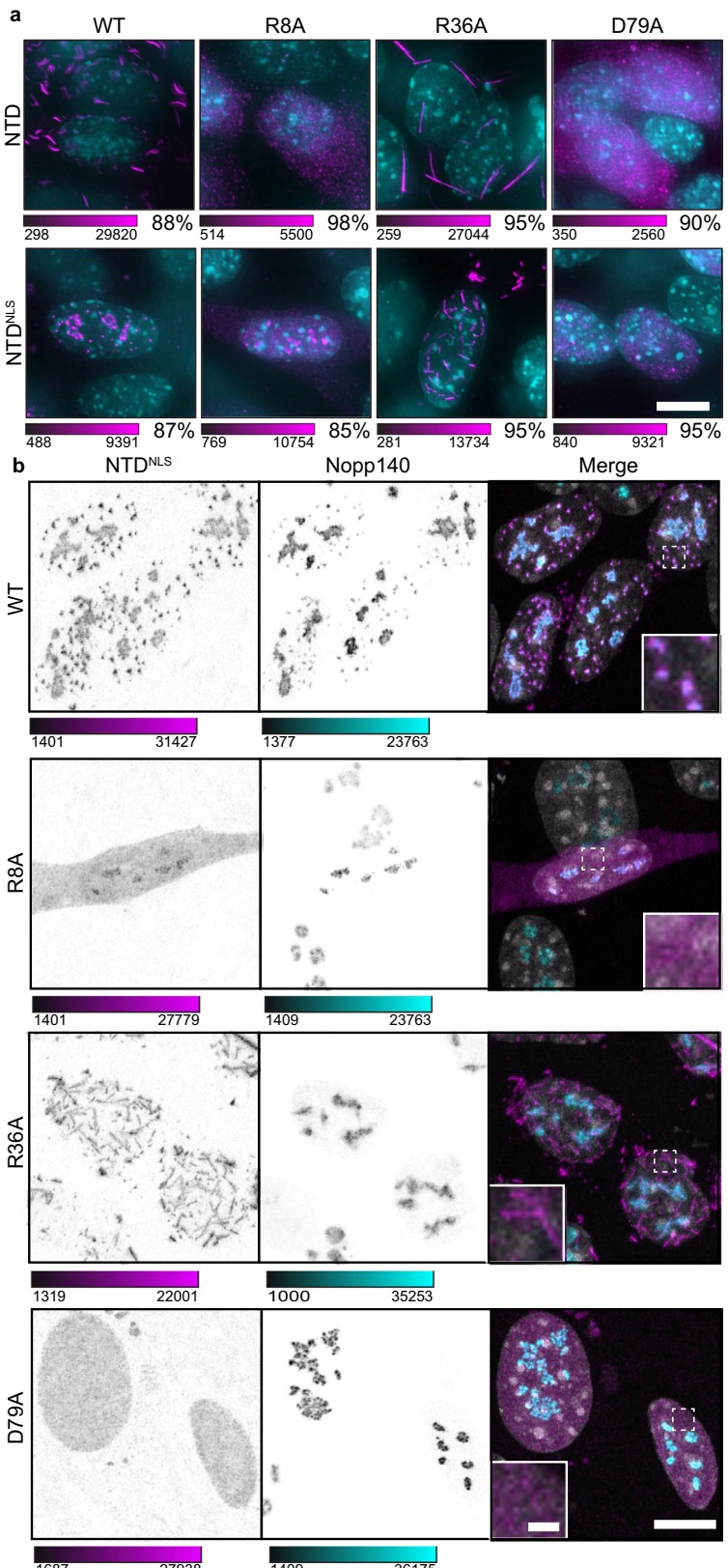

**Fig. 5 | Point mutations alter NTD capacity to form fibrils and puncta. a** *Coil*⁻/⁻ MEFs transfected with constructs for cytoplasmic expression (NTD) or for nuclear expression (NTD^NLS) of the coilin NTD bearing selected point mutations. The NTD was immunostained with anti-myc (magenta) and DNA visualized with DAPI (cyan). Color scale bars given in analog-digital units. Scale bar = 10 μm. Percentage to the lower right indicates the fraction of cells with the displayed phenotype where *n* = 300 cells. Images acquired with DeltaVision. **b** *Coil*⁻/⁻ MEFs transfected with wild-type and mutant coilin nuclear expression constructs (NTD^NLS). NTD was immunostained with anti-myc (magenta), counterstained with anti-Nopp140 (cyan), and DNA visualized with Hoechst (gray). The dashed boxes have been magnified in the inset panels. Imaged with Leica SP8 laser scanning confocal microscope. Color bars are in analog-digital units to indicate differences in expression. Scale bar = 10 μm; inset, 1 μm.

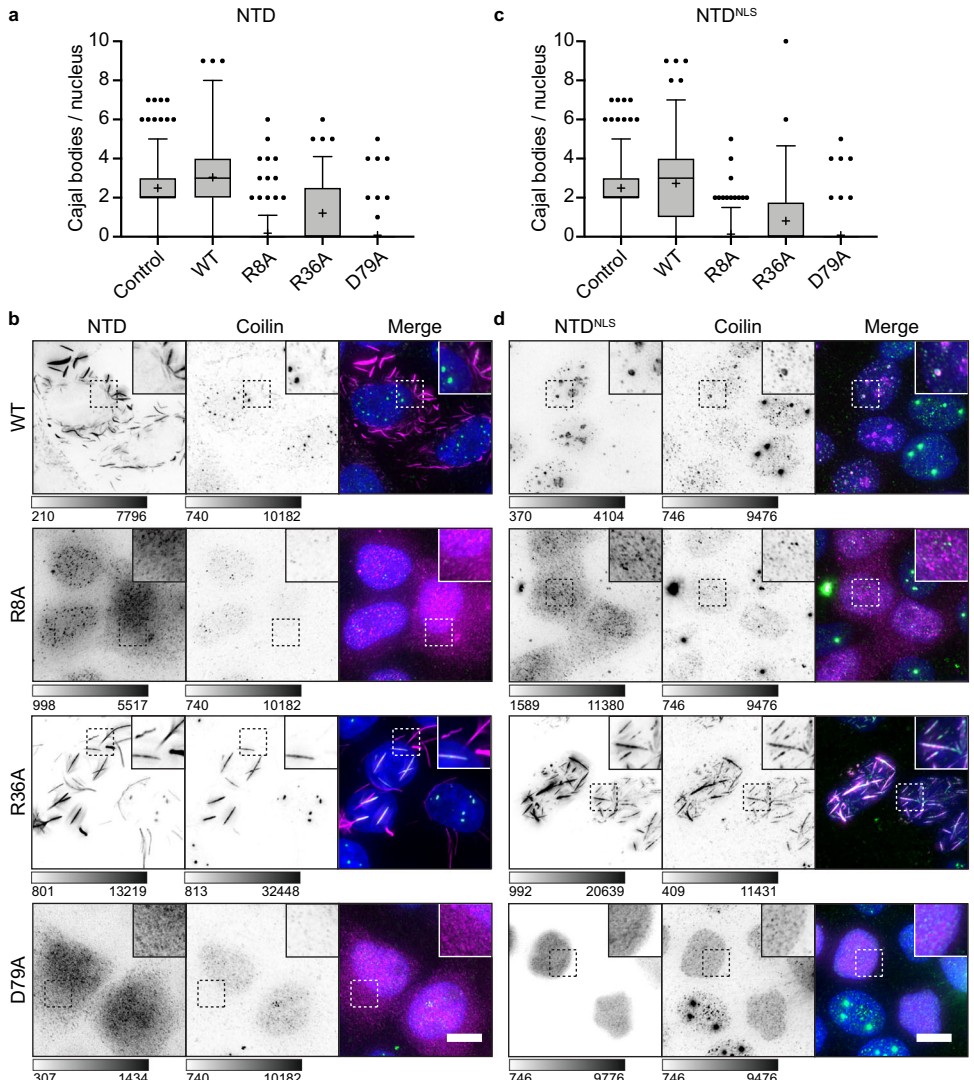

**Fig. 6 | Coilin NTD mutants exert dominant negative effects on endogenous coilin and Cajal bodies.** Wild-type and mutant myc-tagged coilin NTD constructs expressed in the cytoplasm (NTD, **a**, **b**) and nucleus (NTD^NLS, **c**, **d**) of HeLa cells. (**a**, **c**) Cells were immunostained for NTD-myc and endogenous coilin and Cajal bodies were quantified. Measures of center include the median (horizontal line) and mean (cross). The box is drawn from the 25th to the 75th percentile. The whiskers are drawn from the 5th percentile to the 95th percentiles. **a** n = 300 control, 93 WT, 277 R8A, 97 R36A, 51 D40A, and 302 D79A cells measured. **c** n = 300 control, 205 WT, 249 R8A, 48 R36A, 169 D40A, and 299 D79A cells measured. **b**, **d** Representative images show myc (magenta), coilin (green), and DAPI (blue). Grayscale bars given in analog-digital units. Scale bars = 10 μm. Images acquired with DeltaVision. Source data are provided as a Source Data file.

to form higher-order multimers, effectively titrating away free coilin; this phenomenon could be similar to the competition of active G3BP molecules by partially inactive cap constructs in a recent study of stress granule formation[49]. Alternatively, interspersal of R8A and D79A mutant NTDs within naturally occurring coilin fibrils that are components of pre-existing CBs could break them apart, causing CB disassembly. In these two scenarios, we reasoned that R8 and D79 would need to represent two distinct interaction sites: when R8A is present, D79 can still mediate WT coilin interactions with the NTD and vice versa. To test this possibility, we generated a double mutant NTD^NLS harboring both R8A and D79A. If these mutations block two independent sites, then transfection of the double mutant should not affect endogenous CBs. Strikingly, the overexpressed NTD^NLS R8A/ D79A double mutant was diffusely distributed and did not disrupt endogenous CBs (Supplementary Fig. 7a, b), unlike the single mutants. Taken together, the dominant effects of expression of mutant NTDs on endogenous coilin and CBs argue strongly for specific multivalent interactions between coilin NTD–NTD and coilin NTD–Nopp140 at the indicated sites.

## Discussion

This study approaches the Cajal body as an endogenous cellular system, exploiting its conservation over millennia, to expose the molecular details that drive its assembly and shape. The data enable us to transition from the knowledge that coilin is essential for the assembly and maintenance of CBs to a molecular model. Specifically, we provide evidence that the coilin NTD contains at least two distinct faces on the predicted ubiquitin-like fold that mediate coilin–coilin multimerization with the ability to form fibrils. The ability of endogenous coilin to join these fibrils shows that the fibril-forming activity of the NTD is shared with the full-length, wild type protein. These coilin multimers or fibrils may be remodeled into puncta via multivalent interaction with Nopp140. Our working model (Fig. 7) includes the inference that Nopp140 binds coilin multimers, potentially changing their shape from fibrillar to punctate. Another possibility is that Nopp140 interactions sequester smaller coilin assemblies that undergo condensation without forming fibrils.

Our working model (Fig. 7) is supported by in vivo FRET and co-immunoprecipitation data linking morphological phenotypes—

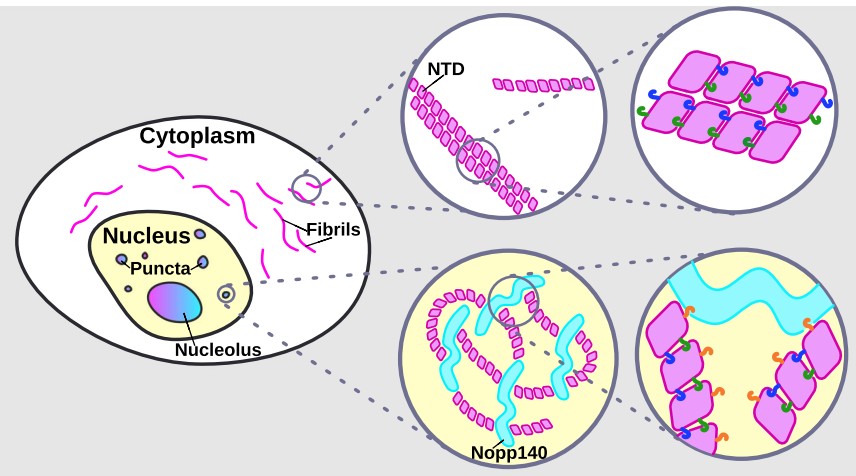

**Fig. 7 | Working model for Cajal body assembly and maintenance via multi-valent coilin NTD–NTD and NTD–Nopp140 interactions.** Schematic depicts coilin NTD fibrils in the cytoplasm, where Nopp140 is absent. In the nucleus, puncta may form by the remodeling of coilin NTD fibrils by Nopp140. Alternatively, biomolecular condensation by Nopp140 could limit coilin NTD fibril formation. Coilin NTD molecules are represented as magenta diamonds and the intrinsically disordered Nopp140 molecules are represented as cyan ribbons. Green, blue, and orange hooks represent NTD residues R8, D79, and R36, respectively.

whether diffuse, fibril, or punctate—to loss of specific interactions due to single amino acid mutations. For example, coilin interaction with Nopp140 is lost when R36 is mutated to alanine, and coilin is observed as long, nuclear fibrils. Consistent with this, wild-type coilin NTD forms fibrils in the cytoplasm where Nopp140 is absent. Conversely, coilin multimerization is blocked when R8 or D79 are mutated, and neither fibrils nor nuclear bodies form. Taken together, the data indicate that both coilin multimerization mediated by R8 and D79 and interaction with Nopp140 mediated by R36 are required for CB assembly. It is unlikely that these mutations disrupt NTD folding, for several reasons: First, the predicted structure places the affected amino acids on the outside surface of the ubiquitin-like fold. Second, if each amino acid changes were destabilizing, then likely each amino acid change would yield a similar phenotype; instead, the R8A, R36A, and D79A mutations have independent phenotypes, and the R8A/D79A double mutant has the opposite phenotype of R8A and D79A alone (see Supplementary Fig. 7). Third, the fact that R8A and D79A mutant NTDs can interact with WT endogenous, full-length coilin indicates the NTD structure is still functional in this capacity. These structure-function relationships provide molecular insights into coilin multivalency.

Our findings highlight the importance of multimerization in the formation of healthy and diseased structures within cells[50–52]. Previous work has detected the formation of aggregates, oligomers, and fibrils in the context of droplet or hydrogel formation. For example, hnRNP A2 adopts a cross-β structure whether it is in hydrogels, droplets or in nuclei[53]. However, coilin fibrils do not appear to exhibit properties of amyloids, because they do not stain with Thioflavin T or Congo Red; moreover, they were negative for ProteoStat staining typical for protein aggregates (see Fig. 1e and Supplementary Fig. 1e). Fibril formation by other proteins, like NPM1 in nucleoli and SPOC in speckles, may underlie scaffolding or localization to those respective structures[54,55]. In our study, we found that the coilin NTD could not be replaced by the GCN4 heterologous dimerization domain, indicating that higher-order multimers and/or additional NTD functions are required for CB assembly. Moreover, coilin itself may not undergo biomolecular condensation in vivo, since coilin^ΔNTD did not form condensates upon blue light-induced dimerization in the optodroplet assay. Instead, our data and resulting model clarify an additional, essential role that Nopp140 plays in the biomolecular condensation of the CB. First, we identified the NTD as the interaction site for Nopp140 and furthermore validate its requirement in CB assembly by depleting Nopp140 and observing CB disruption. Second, we implicated biomolecular condensation by

Nopp140 in CB assembly, by showing that Nopp140's IDR is active in the optodroplet assay and the requirement for Nopp140 for CB assembly and maintenance in cells. Interestingly, Nopp140 and coilin are present in amounts that are consistent with our model. Cellular concentrations of coilin have been calculated to be 101 nM and of Nopp140 to be 60 nM[56], suggesting there are potentially multiple coilin molecules per Nopp140. We propose a model in which coilin oligomers act in trans with Nopp140 to form a minimal unit for assembly of the CB as we have seen in the formation of nuclear puncta formed by NTD^NLS protein.

What, then, is the role of the rest of the coilin molecule? The RG box within the coilin IDR is modified by dimethylarginine required for SMN binding and the docking of gems to CBs[30,32]. The binding of Sm proteins to the coilin CTD likely facilitates snRNP localization and maturation, linking CB assembly to its essential function[35]. The sites of interaction for additional coilin-binding proteins, such as the U6 snRNP component SART3 and the capping enzyme TGS1, as well as coilin-binding snRNAs and snoRNAs have not yet been mapped to particular domains on coilin[3,43]. Despite this incomplete information, the role of the coilin IDR could be simply to link the CB assembly function of the coilin NTD with the snRNP maturation function of the coilin CTD.

Our data reveal specific multivalent interactions of the coilin NTD that shed new light on CB assembly, provoking an evaluation of the role of biomolecular condensation in this process. In 2005, Gall and colleagues speculated that CBs are phase separated based on observed shape, permeability, and differential protein concentrations between the nucleoplasm and the large CBs (10 μm in diameter) present in *Xenopus* germinal vesicles[57]. Note that these coilin-containing spheres may be more closely related to histone locus bodies and are mostly extra-chromosomal[1,58,59]. Subsequent publications have referred to CBs as liquids[46]. Interestingly, electron microscopy (EM) of CBs in somatic cells shows what appear to be coiled electron dense fibers that led to the naming of "coiled bodies" before they were rechristened "Cajal bodies" in honor of their first observer[1,60,61]. Immunogold localization of coilin seems to decorate these coiled fibers when viewed by EM[26]. We wonder if these could represent the fibrils we have observed. Given that filamentous structures can undergo liquid–liquid phase separation, this coiled appearance or the presence of fibrils does not exclude the possibility that the CB is a liquid[62]. Recent sub-diffraction fluorescence imaging has also shown that CB morphology includes indentations or pockets to which other nuclear MLOs, such as gems,

can be docked[32]. How coilin multimers accommodate or facilitate these morphological relationships remains to be determined.

How do CBs achieve their dynamic character? We speculate that disassembly and reassembly—for example, before and after mitosis— may be mediated by the shortening or lengthening of coilin fibrils. Alternatively, we speculate that Nopp140 may associate or dissociate from coilin molecules, e.g., based on phosphorylation state[41]. Although there is no a priori reason to suspect that multimerization of the coilin NTD would be altered by cell cycle, coilin is phosphorylated and methylated at several sites suggesting that post-translational modifications of coilin and its interaction partners might contribute to these changes[36,63]. Moreover, it seems likely that the increase in Nopp140 solubility during mitosis, which is caused by phosphorylation of ~80 serines by casein kinase 2, may play a role in CB dynamics[41,64,65]. Nopp140 is a shared component between CBs and nucleoli, and nucleoli also disassemble at mitosis. Inevitably, these changes must also be linked to the dependency of nucleoli and CBs on transcription, which is required for the presence of these MLOs at active rDNA and snRNA genes, respectively[11,51,66]. Thus, the role(s) of transcription, RNA, post-translational modifications and additional protein interaction partners are leading towards a molecular model for CB dynamics. The present discovery that multimerization by coilin and association with Nopp140 are required for the limited assembly of an approximately spherical membraneless structure suggests a core CB particle that provides the basis for recruitment of the full complexity of CB components.

## Methods

### Experimental model

**Tissue culture cell lines.** The *coil*[−/−] mouse embryonic fibroblasts were a gift from Greg Matera[17]. HeLa cells are from the Kyoto linage (RRID: CVCL_1922). Cry2 optodroplet experiments were performed in NIH-3T3 cells obtained from ATCC (# CRL-1658). 293FT cells were used as a lentivirus packaging line only and were obtained from Thermo (#R70007). All cell lines were cultured in DMEM supplemented with 10% FBS, penicillin, and streptomycin (Gibco) at 37 °C in a 5% $CO_2$ atmosphere. Cells were regularly screened for the presence of mycoplasma infection.

### Method details

**Plasmid and viral vector construction.** Coilin and Nopp140 constructs were cloned into the pEGFP-N1 backbone modified to include a C-terminal c-myc or HA epitope tag, rather than a fluorescent protein. Coilin and Nopp140 constructs used for FRET assays were cloned into pECFP-N1, pECFP-C1, pEYFP-N1, or pEYFP-C1 depending on the measurement. Cry2 constructs were inserted into the pHR_SFFV backbone originally received as a gift from Cliff Brangwynne[47]. Lentiviral particles were generated by co-transfecting the pHR_SFFV vector with pCMV 8.74 (generated by Didier Trono, Addgene #22036) and pMD2.G (generated by Didier Trono, Addgene #12259) into 293FT cells using Fugene HD transfection reagent (Promega). Viral particles were harvested by removing supernatant and filtering through 0.45 μm syringe filters to remove debris.

**Mammalian cell expression and imaging sample preparation.** Transient transfections were performed using Lipofectamine 3000 reagent (Invitrogen) in cells grown on No. 1.5 coverslips (Zeiss). Cells were fixed in fresh 4% paraformaldehyde (Sigma) 24 h post-transfection. Each sample was then blocked and permeabilized in 3% Bovine Serum Albumin (Sigma) and 0.1% Triton X-100 (American Bioanalytical). Immunofluorescence labeling was performed in the same blocking buffer using sequential primary antibody followed by secondary antibody labeled with fluorophore. Nuclei were stained with 1 μg/mL Hoechst 33342 (Invitrogen). Coverslips were then mounted using DABCO.

For live-cell imaging of Cry2 constructs, stable NIH-3T3 cell lines were generated by applying virus to wild-type cells. Cells were then plated onto glass bottomed 35 mm dishes (MatTek). The media was replaced with Live Cell Imaging Solution supplemented with 20 mM glucose and 1 μg/mL Hoechst. For indirect FRET assays, cells were transfected with pairs of CFP and YFP constructs using the conditions described above, fixed in 2% paraformaldehyde, and mounted in DABCO without Hoechst.

Primary antibodies used for imaging include c-Myc antibody (Santa Cruz sc-789; 1:200), Anti-Nucleophosmin antibody (Abcam ab10530; 1:1000), Anti-SMN/Gemin 1 antibody (Abcam ab5831; 1:200), Anti-coilin antibody (Abcam ab210785; 1:300), c-Myc antibody (Santa Cruz sc-40; 1:200), Anti-Nopp140 antibody (Santa Cruz sc-374033; 1:500), and Anti-actin antibody (Sigma Aldrich A4700; 1:200). The secondary antibodies used include Cy™5 AffiniPure Goat Anti-Rabbit IgG (Jackson ImmunoResearch 111-175-144; 1:500), Alexa Fluor 594-conjugated AffiniPure Donkey Anti-rabbit IgG (Jackson ImmunoResearch 711-545-152; 1:200), and Alexa Fluor 488-conjugated AffiniPure Donkey Anti-Mouse IgG (Jackson ImmunoResearch 715-545-150; 1:200).

**siRNA silencing.** Coilin and Nopp140 were depleted using pools of Ambion Silencer Select oligos (ThermoFisher). siRNAs targeting *Coil* (s15662, s15663, and s15664) and *Nolc1* (s17632, s17633, and s17634) were prepared according to the manufacturer's protocols and combined into pools of equimolar amounts of each of the three oligos. A total of 10 pmol were transfected into HeLa cells grown to 50% confluency in a 12-well plate with Lipofectamine RNAiMAX (Invitrogen). After 24 h, cells were re-plated onto coverslips for imaging or for western blot analysis and harvested or fixed after 48 h post-transfection.

**Microscopy platforms.** Fixed cell samples were imaged on one of two platforms as noted in the figure legends. For those imaged using the DeltaVision platform (Applied Precision), a 60× PlanApo N.A. 1.42 Oil objective (Olympus) was used to image the full depth of the sample at 0.2 μm intervals. The resulting wide field image stacks were deconvolved with the native Applied Precision software. For samples imaged using laser scanning confocal, a Leica Sp8 was used with a 63× HC PL APO CS2 Leica objective. The instrument is equipped with a white light laser, 775 nm STED (stimulated emission depletion) laser, three hybrid detectors, and two PMT detectors for spectral selection, and channels were imaged sequentially to eliminate bleed through. Confocal stacks were deconvolved using the Huygens Professional software. All images are displayed as maximum intensity projections. For STED, Images acquired below the diffraction limit were collected using STED to achieve a lateral resolution of approximately 50 nm.

Live-cell imaging of Cry2 samples was performed on a Bruker Opterra II Swept Field Instrument. Samples were simultaneously illuminated with 488 and 561 nm laser light, where the 488 nm light activates Cry2 and the sample was imaged in the 561 nm channel to detect mCherry. A PlanApo 60 × 1.2 NA water immersion objective was used and the instrument is equipped with a Evolve 512 Delta EMCCD camera. Ten frames at one second each were captured before activation for a total of 180 s. A stage-top incubator was used to maintain cells at 37° throughout the imaging protocol.

**FRET measurements.** Flourescence resonance energy transfer (FRET) measurements were made using the acceptor photobleaching scheme[67]. Measurements were made on a Zeiss 710 laser scanning confocal instrument equipped with 458 and 514 nm lasers for imaging CFP and YFP respectively, detected by corresponding spectrally selective PMT detectors. Measurements were made by imaging both channels in a given region of interest (ROI) followed by a high intensity bleaching scan of the ROI with the 514 nm laser. The same ROI was then

reimaged in both channels. A fusion construct of CFP-YFP was used as a positive control and freely expressed non-fused CFP and YFP were used as a negative control.

**Immunoprecipitation assays and western blotting.** Immunoprecipitation assays were performed by transfecting a bait GFP fusion protein into HeLa cells as discussed above. Approximately eight million cells were harvested for each sample and lysed into Pierce lysis buffer (Thermo) containing Complete protease inhibitors (Roche). Lysate was clarified by centrifugation at $10,000 \times g$ for 30 min. Pellet and a portion of supernatant were reserved as input. The remaining lysate was incubated with 4 µg of goat anti-GFP overnight, followed by immunoprecipitation with Protein G Magnetic SureBeads (BioRad). Beads were washed three times in Pierce lysis buffer and eluted into NuPAGE Sample Buffer (Invitrogen). The resulting eluent was run onto a NuPAGE 4–12%, Bis-Tris gel (Thermo), transferred to nitrocellulose and probed with primary antibody to be visualized with horseradish peroxidase conjugated secondary antibody. The antibodies used were NOLC1 antibody (NovusBio NBP1-2298), GFP antibody (Invitrogen A-11122), and anti-rabbit IgG Horseradish Peroxidase-Linked Species-Specific Whole Antibody (GE HealthCare NA934).

**Amyloid dye staining.** Amyloid dye samples were prepared for microscopy as described above for immune fluorescence. Following antibody probing with Cy5 to prevent bleed through, samples were stained with either Thioflavin T (200 nM, Sigma), Congo Red (0.005%, Sigma), or Proteostat (3 µM, Biotium) solutions.

### Quantification and statistical analysis

**Indirect FRET analysis.** Indirect FRET was carried out in fixed cells expressing the specified CFP and YFP FRET pairs and as described previously[4,13,67]. The bleached portion of each image was cropped to a $30 \times 30$ pixel ROI. Apparent FRET efficiency is expressed as a percentage for that ROI, calculated as $E_f = (I_2 - I_1)/(I_2)$ where $I_2$ is the mean intensity of the CFP channel after photobleaching of YFP, and $I_1$ is the intensity of the CFP channel before photobleaching. $I_1$ and $I_2$ were background corrected by subtracting the mean intensity of five ROIs outside of cells. The reported FRET efficiency is the mean of twelve ROIs from two separate biological replicates with error bars as standard error of the mean.

**Automated quantification of Cajal bodies.** Cajal bodies per nucleus were counted by segmenting nuclei from images of Hoechst staining and coilin from images of coilin immunofluorescence using minimum cross-entropy thresholding and a three-class foreground Otsu threshold, respectively. The identified coilin puncta were then filtered to only count those occupying more than 4 pixels, removing those objects under 0.5 µm in diameter.

**Relative saturation concentration estimation.** Saturation concentration was estimated by comparing the percentage of the nucleus occupied by condensates to the expression level of the construct in that nucleus. This parameter was measured in two dimensions by determining the pixel area occupied by each nucleus estimated by thresholding maximum intensity projections of Hoechst staining using minimum cross-entropy. The area occupied by condensates was determined by applying a fixed threshold above which all pixels were assumed to be condensates. The fixed threshold was chosen by manually inspecting the positive control condition (full length coilin or wild-type coilin NTD) and setting it just below the intensity of the lowest expressing cells that still form condensates. A percentage of condensed nuclear space was thus determined by comparing this number to the total nuclear area. The expression level of a construct was estimated by generating a summed projection from three dimensional stacks and computing a mean-intensity of each nucleus to normalize for nuclear size.

**Manual nuclear body and coilin mutant phenotype quantification.** To complement automated quantification of nuclear bodies, cells were counted by hand. At least two biological replicates per condition were analyzed. An author previously unfamiliar with the data was asked to establish phenotype categories and count the number of positively transfected cells matching those categories. Likewise, they counted the number of nuclear bodies per nucleus in conditions where nuclear bodies were quantified. The sample size is noted in the respective figure legends.

**Image display.** All images within subfigures are acquired on the same instrument using the same settings, and the platform used is specified in the figure legend. Where images are compared, a grayscale or color bar is provided to indicate the intensity value range of the image. If images have not been intensity matched, multiple color bars are provided to indicate the range for each image.

**Software.** Images were prepared for figures using ImageJ (FIJI) version 2.0.0-rc-69/1.52p[68]. Structure prediction was performed on the Jpred and RaptorX server using default parameters[48,69]. Coilin alignments were performed using Clustal Omega using the default parameters[70]. Automated microscopy analysis was carried out with Cell Profiler 4.2.1[71].

**Statistics and reproducibility.** All experiments displayed in the figures and supplement were repeated three times with similar results, unless otherwise specified in the figure legend. Note that the acceptor photobleaching FRET assay where apparent FRET was calculated in 12 cells in each of two replicates. The results were highly reproducible.

### Reporting summary

Further information on research design is available in the Nature Research Reporting Summary linked to this article.

## Data availability

This study did not generate large datasets. Source data are provided with this paper.

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

## Acknowledgements

The authors wish to thank members of the Neugebauer lab for helpful discussions and comments on the manuscript. We thank the Yale Center for Advanced Light Microscopy Facility for their assistance with confocal microscopy. We are grateful to Daniela Šimčiková and Petr Šimčik for preparing the illustration in Fig. 7. We thank Greg Matera for the *Coil–/–* MEF cell line and for helpful discussions. The study was supported by NIH awards NINDS-F31NS105379 (to E.M.C.), NINDS-R01NS128358-01 (to K.M.N.), and U01CA200147 TCPA-2017-Neugebauer (to K.M.N.). J.E. was supported by the Yale BioMed Amgen Scholars Program through a grant from the Amgen Foundation (41716881). This work is solely the responsibility of the authors and does not necessarily represent the official views of the NIH.

## Author contributions

M.M., E.M.C., and K.M.N. designed the study. M.M., S.S., and K.S. generated all constructs. M.M., E.M.C., S.G.W., S.S., J.E., and K.S. prepared cell lines and performed experiments. E.M.C., S.G.W., and J.E. carried out image analysis. K.M.N. supervised the study. K.M.N., E.M.C., and S.G.W. wrote the manuscript and all authors contributed comments and edits.

## Competing interests

The authors declare no competing interests.
