## [Peer Review File · Nature Communications]

The coilin N-terminus mediates multivalent interactions between coilin and Nopp140 to form and maintain Cajal bodiesREVIEWER COMMENTS

Reviewer #1 (Remarks to the Author):

Courchaine et al. investigate the function of coilin and Nopp140 in Cajal body (CB) formation. For this purpose, they mainly rely on coilin $-/-$ MEFs (with only residual CBs) to test the induction of CBs by transfection of coilin constructs. The focus is on the N terminal domain (NTD), which was previously identified as self-interaction domain required for CB localization. When expressed alone, the NTD localizes to remarkable $10 \times 0.1 \mu\text{m}$ filaments (referred to as oligomers) in the cytoplasm. In contrast, the addition of an SV40 nuclear localization signal (NLS) to the NTD (NTD-NLS) now targets the NTD-NLS to nuclei, where it localizes in several foci of various size. Interestingly, the localization of transfected NTD-NLS matches that of endogenous Nopp140. Experiments with siRNAs against Nopp140 and coilin in HeLa cells (with CBs) shows a reduction by less than half and a little more than half of the number of CBs, respectively. Foci of SMN are only affected by Nopp140 but not coilin silencing. Through an alanine scan, 3 amino acids in the conserved NTD are identified, R8A, R36A, and D79A, that impact the localization of the NTD when transfected into coilin $-/-$ MEFs. Coilin-coilin and coilin-Nopp140 FRET in HeLa cells shows R8A to abolish both interactions, whereas R36A only reduces the coilin-Nopp140 FRET. These experiments are substantiated by co-immunoprecipitations of GFP-tagged wild type and mutant coilin. Structure prediction of the ubiquitin-type fold of the NTD reveals a barrel-like shape formed by 4 beta-sheets and one alpha-helix with the 3 mutated amino acids adorning different surfaces of the barrel. Transfection of wild type and mutant NTD into coilin $-/-$ MEFs shows the cytoplasmic filament formation to be abolished by the R8A and D79A mutations, whereas the R36A mutation exaggerates filament formation in the cytoplasm. A similar result is obtained after addition of an NLS to the NTD, except that R36A now forms filaments also in the nucleoplasm. Similar results are seen in a dominant negative-type fashion when the same constructs are transfected into HeLa cells where they reduce the number of CBs but R36A still forms filaments. Fusing the Cry2 domain, which dimerizes when illuminated, to full-length coilin induces additional CBs when light activated. But the construct remains completely diffuse if the NTD is removed. When fused to part of the Nopp140 intrinsically disordered domain, light activation causes the normally diffuse construct to match the localization of endogenous Nopp140 in nucleoli and CBs. Interestingly, siRNA treatment of Nopp140 causes transfected NTD-NLS to form filaments in nucleoli, which also contain endogenous coilin. Based on these and additional data, the authors conclude that the NTD is required for oligomerization of coilin and association with Nopp140 to remodel the oligomers and form CBs.

Sorry for the lengthy summary, but I believe it is necessary to appreciate the review and to convey some of the complexity of the study. Overall, these are interesting and exciting findings that should be of interest to a general audience if sorted out properly. The study has a few major flaws and many minor ones that all need addressing. There are some serious overstatements and a potential misinterpretation of the main results, the titular "oligomerization" of coilin.

1. The coilin "oligomers" may have been misidentified as such for the following reasons.

a) There is a problem with the size of the "oligomers". Oligomers normally consist of 3 > 100 repeating units. However, what is observed are filaments of 10,000nm in length with a 100nm diameter, which would easily accommodate 50,000 molecules with a ubiquitin-type fold like the NTD. This is a very conservative estimate assuming a 10 nm diameter for the ubiquitin-type fold, which is apparently even below 2 nm (G-actin, which is about 4-times the size, is 4-7 nm). Consequently, it is very unlikely that those interesting structures are made up of purely NTD or coilin. The very least, those structures are polymers of substantial size, i.e., filaments.

b) Based on the form and shape, I suggest that these structures are actin-based filaments (decorated by NTD). Electron microscopy or simple staining with phalloidin could easily resolve this question. Of course, they could also be other types of cytoskeletal filaments.

c) Actin filaments of that size are present in the nucleoplasm and have been known to be induced in nuclei of stressed cells in culture for over 40 years. Among other stresses, heat shock and DMSO have been documented to induce intranuclear actin rods. Here are a few references, which even show actin rods in nucleoli as observed in this study (Fig. S5D), where the knockdown of Nopp140 by siRNAs might constitute a stress.

Fukui, Y. (1978). Intranuclear actin bundles induced by dimethyl sulfoxide in interphase nucleus of *Dictyostelium*. *J Cell Biology* 76, 146–157.

Iida, K., Iida, H., and Yahara, I. (1986). Heat shock induction of intranuclear actin rods in cultured mammalian cells. *Exp Cell Res* 165, 207–215.

Amankwah, K.S., and De Boni, U. (1994). Ultrastructural Localization of Filamentous Actin within Neuronal Interphase Nuclei in Situ. *Exp Cell Res* 210, 315–325.

d) There are some additional observations and questions pertaining to point 1.

The filaments seem to be polarized, like actin comets in the case of *listeria*, perhaps indicating a preference of the NTD for the polymerizing or depolymerizing end of actin?

Why are the filaments not formed in the context of full-length coilin, i.e., in the R36A mutant, which also goes to the nucleus (compare Fig. 3B and 4B)?

Could the point mutations affect the structure of the ubiquitin fold itself rather than its interaction?

Even the nuclear puncta are sufficiently large to harbor polymers, not only oligomers.

2. Most of the experiments are performed in coilin ^{-/-} MEFs known for their lack of normal CBs, but with “residual CBs”. In those cells, coilin is only deleted from the second exon onwards, such that the NTD alone could still be expressed. In fact, the authors from that study were so concerned with that possibility that they exogenously expressed a GFP-tagged NTD (GFP-mcoilin^{KO}), which localized to CBs in wild type MEFs but not in coilin ^{-/-} MEFs. Interestingly, close inspection of its expression in the coilin knockout MEFs shows something like filaments in the cytoplasm (Fig. 7A of that study). Regardless, it is a serious problem if the NTD were already expressed in the cells used for most of the studies. This point needs to be clarified.

3. It is questionable to study membrane less organelles (MLOs), which are extremely sensitive to variation in concentration of their components, by exogenous expression of some of the components. Obviously, the mere expression of coilin or NTD will affect CBs raising questions about the reported observations.

4. As interesting and unexpected the NTD filaments may be, they are not physiological but artificially induced. This does not mean that they are not a worthwhile phenomenon to study or learn from but lessens the general interest.

5. Although the link between Nopp140 and coilin is interesting, there are several holes in that story.

a) NTD-NLS and Nopp140 colocalize exactly in nucleoli and CBs. This is not observed for the NTD alone, which presumably stays anchored in the cytoplasmic filaments despite being small enough to diffuse into the nucleus. Rather than the NTD interacting with Nopp140, it is equally possible that it is the NLS part that interacts with Nopp140. In fact, the very same SV40 NLS was used to identify and purify Nopp140 and shown to colocalize with Nopp140 upon incubation with permeabilized cells. If this is what is observed in this study, the interpretation of the results is wrong.

b) The optodroplets of the Nopp140 intrinsically disordered region (IDR) induced by light in nucleoli and CBs are quite interesting (Fig. 6C). However, the interpretation is illogical. Before light induced dimerization, the Nopp140 IDR is diffusely localized throughout the nucleoplasm but after light activation, it matches the localization of endogenous Nopp140 in nucleoli and CBs. This suggests that

dimerization of the Nopp140 IDR is required for its association with endogenous Nopp140, not with coilin, which is only in CBs.

c) Fig. 2 claims that Nopp140 is essential for proper CB assembly. All CB components are required for "proper" CB assembly. For example, it was published that knockdown of Nopp140 in stable cell lines causes a displacement of all scaRNPs from CBs leaving behind coilin, snRNPs, and SMN in condensates that are no longer "proper" CBs. Regardless, the knockdown of coilin and Nopp140 (Fig. 2) by transient transfection of siRNAs, where differential effects on coilin and SMN puncta are observed, needs to be better controlled. For example, in immunofluorescence panels E and F, it is not clear which cells are silenced, partially silenced, or not at all silenced. To determine the effect on Nopp140, a Nopp140 stain is required. In panel F, the line profile of a cell that is not silenced at all seems to be shown claiming increased coilin in the nucleoplasm after Nopp140 targeting. Obviously, this needs to be controlled. In the quantification panels B and C, only positively identified silenced cells should be counted.

d) Nopp140 is only targeted by siRNAs in cells with endogenous coilin. Would CBs still form if coilin were transfected into coilin $-/-$ MEFs after Nopp140 silencing?

e) As already stated, NTD-NLS colocalizes with Nopp140, e.g., Fig. 1F. Then why is the NTD-NLS in Fig. 1C only in puncta and not in nucleoli? Also, Fig. 1F is shown twice, again in Fig. S1E upper panels. Are there not sufficient images to show different nuclei with the same result? Does the localization vary?

6. There appear to be some inconsistencies with the FRET data. The FRET efficiency of full length coilin is about two-fold higher when the FRET labels are at its N versus C termini, and close to zero when put on opposite ends. When the NTD is replaced by the GCN4 dimerization domain, the N terminal labels still yield FRET, but the C terminal ones result in negative FRET, clearly identifying a difference in behavior between the NTD and GCN4. Therefore, the data shows the opposite of "confirming the functionality of the GCN4 dimerization domain (Fig. S1b)" (page 6). Hence, and counter to what is claimed, it is not surprising that GCN4-coilin behaves differently from coilin with the NTD when transfected. Similar misleading setups are used on other occasions in the manuscript. Moreover, since FRET seems to be a bit capricious, it would be important to show permutations with N and C terminal labels for the Nopp140 coilin FRET, i.e., as for coilin in Fig. S1B. Obviously, the FRET results must be taken with a grain of salt.

7. The dominant negative effects of the NTD constructs in HeLa cells with coilin and CBs are interesting (Fig. 5). Not described is the effect of the wild type NTD-NLS, which colocalizes with Nopp140 in nucleoli and CBs and which disperses endogenous coilin from CBs (Fig. 5D). The data are quite clear as three transfected cells (on the right) and three untransfected cells (on the left) are shown. Perhaps the transfected or the untransfected cells should be labeled in those panels. Anyway, does this mean that coilin is only held in CBs through its NTD? An interesting result that should be pointed out.

In summary, the authors approach a complex problem and try to put an interesting spin on it. However, the manuscript must have gone through many iterations, as figures are mislabeled (by small letters in the text but capitals in the figures) or altogether not referred to (Fig. S3B). Regarding S3B, what does it mean "soluble" versus "insoluble", does that draw the immunoprecipitations in Fig. 3E into question? Reviewing was not made easier by the lack of descriptions of material and methods, e.g., none of the antibodies are described nor is STET. After a thorough overhaul including proper controls and experiments as pointed out above, I would certainly look forward seeing some of this perplexing data published, but not in the present form.

Reviewer #2 (Remarks to the Author):

In "Coilin oligomerization and remodeling by Nopp140 reveals a hybrid assembly mechanism for Cajal bodies" Courchaine, Machyna et al., focus on elucidating the molecular principles of Cajal body (CB) assembly that have largely remained elusive despite the long standing knowledge of the essentiality for Coilin. They provide some conclusive insights for the importance of both Coilin oligomerization and for the interaction between Coilin and Nopp140 in CB assembly within double knockout coilin MEF cells. Given the well known function of Cajal bodies and well documented importance they have in spliceosome biogenesis their mechanistic insights will be impactful. That said, I have significant concerns regarding the lack of description behind the microscopy data and the often poor quantification of it and the extent to which constructs are expressed. It is essential to understand the extent to which the authors express various constructs to be able to understand the extent to which a given construct can or cannot promote assembly of CBs, CB-like foci, or droplets. Without changes in these two elements, as it stands their data as presented cannot fully support the major elements of their manuscript.

Concerns:

1)The major overarching theme behind my primary concerns with the manuscript pertain to the quantification of the microscopy data. Throughout the text, statements such as X forms or does not form CB/fibers are frequently stated. Representative microscopy images are provided typically with a scale bar for intensities. However, little discussion is mentioned for if and if so how the absolute intensities can be related, even to images which are adjacent. Do the authors always use the same settings for the same microscopes? Alternatively, to increase dynamic range, do they have methods to reference the intensities to one setting (e.g. calibrated function/table to convert from any setting to a reference one)? Having this is important because- as it stands- the readers cannot assess if one construct expressed poorly, and thus precluded CB/fiber assembly or if 10X of expression was needed compared to another to result in the formation of CB/fibers (see concern 2 for more on this). As will be mentioned specifically in multiple locations below the authors need to address how the expression and localization seen via microscopy is quantified in cases where constructs are compared. Otherwise there remains significant doubt that the results aren't dependent on other variables (e.g. microscopy settings or what tag was used).

2)In the field of phase transitions (whether first order, e.g. LLPS, or higher order, e.g. micellization), the essential components of such processes have a concentration whereupon the phase transition occurs often referred to as the saturation concentration or C_{sat} (but also via other names such as the CMC for micelles). This has received significant attention in the field of membraneless organelles//biological condensates (A. Klosin et al., Science 2020; Riback et al., Nature 2020; J. M. Choi et al., PLOS Comput. Biol. 2019). Thus where possible the authors should attempt to determine if there exists a threshold where CB, CB-like, or fibers occur. For example, is Coilin-FL essential for Cajal bodies? If so below some nucleoplasmic C_{sat} there should be no cajal bodies. Similarly when assessing constructs that do not support assembly (e.g. Coilin-deltaNTD) the max overexpression where this is true, relative to a construct that does support assembly such as FL-Coilin, should be reported. If for example Coilin-deltaNTD was expressed only 0.5, 2, or 100 of the nucleoplasmic levels that the FL coilin needed to assemble, this would be no, minimal, high, support for the stated claims that Coilin-deltaNTD cannot drive CB formation. As such, this will allow the readers to assess to what extent the conclusions that various constructs do not support assembly can be assessed. To aid the authors, I will try to mention all the locations where I think this is relevant below.

3)Keeping comments 1&2 in mind. In figure 1B, the amount of overexpression where FL-Coilin forms CBs should be reported and contrasted with the extent to which Coilin-deltaNTD and with the GCN2 dimer were overexpressed. How far did the authors push the GCN2 dimer construct to assess if it was sufficient at high levels to form CBs or CB-like structures.

4)Can the authors comment on the punctate pattern in the nucleoplasm for many coilin constructs? Is

this an artefact of the antibody staining (vs. fluorescent tags). Also it's difficult in many cases to determine what is being imaged (which antibody/fluorescent tag etc). Reporting these in the figure or figure legend would be incredibly useful for the reader.

5)The authors seem confident that the fibers are simply driven by interactions between Coilin-NTD molecules. What is the support for this? One way I could envision demonstrating these are driven by the oligomerization of Coilin is by confirming that there is a fixed Csat for fibrillization. A lack of fixed Csat is indicative of other components driving/impacting the stability of assembly (for example Riback et al., Nature 2020; Posey et al., JBC 2018). Without evidence, the language should probably be toned down accordingly.

6)Could the authors comment on the amount (e.g. Csat) of NTD-NLS needed to form CB-like structures vs. FL needed to form bonafide CBs. Do the other domains in Coilin contribute to stabilizing CBs (i.e. is the Csat nearly the same)?

7)The authors make specific claims about more Coilin being in the nucleoplasm when referencing fig 2d-f. Addressing concern (1) above is fairly important to confirm that these can in fact be compared quantitatively as they intend.

8)With respect to concern (2) above, how is the total overexpression compared between the ALA mutants in figure 3 and corresponding supplement figures. For the ones which do not support CB formation at any concentration what is the max overexpression? Can the authors rule out that some variants just do not simply express weaker than others? Ideally if the authors could compare the Csat for the ALA mutants with WT this would better quantify mutants which may destabilize CB assembly but do not fully eliminate it (e.g. because their mutations may only weakly perturb the unknown interaction surfaces).

9)Figure 4 and 5 needs to be addressed with respect to concerns 1&2 above in terms of the extent to which the constructs are expressed.

10)The authors assert that R8A in figure 4B is not forming CB but going to nucleoli. Can the authors elaborate how they assess this? These puncta look similar to CBs to this reviewer.

11)Can the authors clarify or comment on the result as to why the NTD-R8A which fails to oligomerize and interact with Nopp140 would impact Cajal bodies in Fig 5? Does this suggest another interaction partner?

12)Is the disassembly of Cajal bodies due to truncated Coilin dose dependent similar to the competition assay of Figure 3 in Sanders et al., Cell 2020.

13)It has been observed that overexpression of Coilin results in larger Cajal bodies in HeLa cells and is stabilized by heterotypic interactions between coilin and other components (Riback et al., Nature 2020). While the number does not seem to change as reported in Figure 5 with the overexpression of NTD-NLS. Does the overexpression of Coilin FL and/or NTD-NLS result in increases in size to CBs in these cells?

14)Given the concentration dependence of the Cry2 system for light triggered activation as reported in Shin et al., Cell 2017, it seems important to address the relative overexpression levels (e.g. concerns 1&2 applied to the Cry2 data). Also was Cry2-CoilinDeltaNTD also tested? It seems that this would better mimic the architecture than CoilinDeltaNTD-Cry2. Given the claim that the homotypic interactions between the Nopp140 IDR may be required for CB assembly (due to the results of the Cry2 assay with the Nopp140 IDR) and not simply that Nopp140 yields additional multivalency and prevents fiber-like oligomers, it would seem the correct orientation would better mimic the FL protein.

15)The authors imply that the occurrence of coiled electron dense fibers within CBs by EM as indicative of a lack of liquid nature of CBs. Given that fibers can undergo LLPS into liquids (e.g. Weirich et al., PNAS 2017), presenting the EM as in disagreement with CB as being liquids as opposed to reporting the internal structure of CBs seems inaccurate and misleading.

16)The authors state that 50% of Coilin molecules are immobile and cite reference 11 (Dundr et al., JCB 2004). This is incorrect as reference 11 as done could not assess immobility; the authors of 11 of that work make no such conclusions. As such these authors need to correct this (or correct this reviewer on why they make such claims when the ref 11 authors do not). In fact others (Bártová et al 2014 - <https://doi.org/10.4161/nucl.29229>), report significant recovery after <1min. Furthermore, I don't quite understand why the authors' model requires Coilin oligomers to be immobile. Small oligomers such as NPM1 show >90% recovery on the min timescale (Zhu et al., PNAS 2019). If this is essential, the authors could FRAP CBs and NTD coilin cytoplasmic fibers to validate that Coilin oligomers aren't infact dynamic in these contexts (being sure to correct for the fraction of molecules bleached as bleaching a whole CB often bleaches a significant amount of the total fluorescent Coilin- a problem I think ref 11 was trying to avoid).

17)Minor- Supp figure 4D - Is Sat3-RFP a mistake?

RESPONSE TO REVIEWER COMMENTS

We thank both reviewers for their constructive criticisms, to which we have responded in every case. Many of the issues were resolved by conducting new experiments or repeating previous ones to provide the best data possible in this resubmission. We are grateful to the reviewers for this input and appreciate their patience regarding the timeline. Due to the pandemic, we needed to recruit a new scientist to conduct this experimental work (Sara Gelles-Watnick, now a co-first author). In this substantially revised version, we have added 5 prominent figures to the main text (Figures 1f, 2b-e, 3d, 5b, and 7), including a schematic of our working model. We have added 8 figures to the supplement (Figures S1c&d, S2a&b, S3c, and S6b-e). Please note that we have changed the order of the figures for clarity. In our responses, we are using the new figure numbers to refer to the data. Naturally, all of these changes have necessitated the re-writing of the manuscript accordingly. We hope these measures satisfy the high standards of both reviewers and also clarify the novelty and importance of our work. We look forward to any additional feedback.

In the text below, we have copied each review in its entirety and replied in blue.

Reviewer #1 (Remarks to the Author):

Courchaine et al. investigate the function of coilin and Nopp140 in Cajal body (CB) formation. For this purpose, they mainly rely on coilin $-/-$ MEFs (with only residual CBs) to test the induction of CBs by transfection of coilin constructs. The focus is on the N terminal domain (NTD), which was previously identified as self-interaction domain required for CB localization. When expressed alone, the NTD localizes to remarkable $10 \times 0.1 \mu\text{m}$ filaments (referred to as oligomers) in the cytoplasm. In contrast, the addition of an SV40 nuclear localization signal (NLS) to the NTD (NTD-NLS) now targets the NTD-NLS to nuclei, where it localizes in several puncta of various size. Interestingly, the localization of transfected NTD-NLS matches that of endogenous Nopp140. Experiments with siRNAs against Nopp140 and coilin in HeLa cells (with CBs) shows a reduction by less than half and a little more than half of the number of CBs, respectively. SMN puncta are only affected by Nopp140 but not coilin silencing. Through an alanine scan, 3 amino acids in the conserved NTD are identified, R8A, R36A, and D79A, that impact the localization of the NTD when transfected into coilin $-/-$ MEFs. Coilin-coilin and coilin-Nopp140 FRET in HeLa cells shows R8A to abolish both interactions, whereas R36A only reduces the coilin-Nopp140 FRET. These experiments are substantiated by co-immunoprecipitations of GFP-tagged wild type and mutant coilin. Structure prediction of the ubiquitin-type fold of the NTD reveals a barrel-like shape formed by 4 beta-sheets and one alpha-helix with the 3 mutated amino acids adorning different surfaces of the barrel. Transfection of wild type and mutant NTD into coilin $-/-$ MEFs shows the cytoplasmic filament formation to be abolished by the R8A and D79A mutations, whereas the R36A mutation exaggerates filament formation in the cytoplasm. A similar result is obtained after addition of an NLS to the NTD, except that R36A now forms filaments also in the nucleoplasm. Similar results are seen in a dominant negative-type fashion when the same constructs are transfected into HeLa cells where they reduce the number of CBs but R36A still forms filaments. Fusing the Cry2 domain, which dimerizes when illuminated, to full-length coilin induces additional CBs when light activated. But the construct remains completely diffuse if the NTD is removed. When fused to part of the Nopp140 intrinsically disordered domain, light activation causes the normally diffuse construct to match

the localization of endogenous Nopp140 in nucleoli and CBs. Interestingly, siRNA treatment of Nopp140 causes transfected NTD-NLS to form filaments in nucleoli, which also contain endogenous coilin. Based on these and additional data, the authors conclude that the NTD is required for oligomerization of coilin and association with Nopp140 to remodel the oligomers and form CBs.

Sorry for the lengthy summary, but I believe it is necessary to appreciate the review and to convey some of the complexity of the study. Overall, these are interesting and exciting findings that should be of interest to a general audience if sorted out properly. The study has a few major flaws and many minor ones that all need addressing. There are some serious overstatements and a potential misinterpretation of the main results, the titular “oligomerization” of coilin.

1. The coilin “oligomers” may have been misidentified as such for the following reasons.

We thank the reviewer for this comment and note that the language surrounding extended structures formed by certain proteins is currently controversial and highly discussed by the field. Specifically, whether a visualized subcellular object should be called an oligomer, filament, fibril or any other term can be disputed due to differing assumptions about what these terms imply. We provide arguments below for our choice of terminology.

a) There is a problem with the size of the “oligomers”. Oligomers normally consist of 3 > 100 repeating units. However, what is observed are filaments of 10,000nm in length with a 100nm diameter, which would easily accommodate 50,000 molecules with a ubiquitin-type fold like the NTD. This is a very conservative estimate assuming a 10 nm diameter for the ubiquitin-type fold, which is apparently even below 2 nm (G-actin, which is about 4-times the size, is 4-7 nm). Consequently, it is very unlikely that those interesting structures are made up of purely NTD or coilin. The very least, those structures are polymers of substantial size, i.e., filaments.

We concede that the term oligomers could be confusing based on these arguments. The extended structures we observed are up to 10 micrometers long in the cytoplasm (Fig. 1c). Despite this length, the structures appear to be as little as 100 nm in diameter by sub-diffraction imaging (Fig. 1d). Based on these dimensions, which are similar to collagen fibrils, we have chosen to refer to these structures as fibrils throughout the manuscript. We now explicitly state our rationale in the results section.

b) Based on the form and shape, I suggest that these structures are actin-based filaments (decorated by NTD). Electron microscopy or simple staining with phalloidin could easily resolve this question. Of course, they could also be other types of cytoskeletal filaments.

We have addressed the question of whether the fibrils are actin filaments by staining for actin, as suggested. The data show that actin filaments do not overlap with the extended structures that we now show in the new Figure 1F. We thank the reviewer for suggesting this clarifying experiment.

c) Actin filaments of that size are present in the nucleoplasm and have been known to be

induced in nuclei of stressed cells in culture for over 40 years. Among other stresses, heat shock and DMSO have been documented to induce intranuclear actin rods. Here are a few references, which even show actin rods in nucleoli as observed in this study (Fig. S5D), where the knockdown of Nopp140 by siRNAs might constitute a stress.

Fukui, Y. (1978). Intranuclear actin bundles induced by dimethyl sulfoxide in interphase nucleus of *Dictyostelium*. *J Cell Biology* 76, 146–157.

Iida, K., Iida, H., and Yahara, I. (1986). Heat shock induction of intranuclear actin rods in cultured mammalian cells. *Exp Cell Res* 165, 207–215.

Amankwah, K.S., and De Boni, U. (1994). Ultrastructural Localization of Filamentous Actin within Neuronal Interphase Nuclei in Situ. *Exp Cell Res* 210, 315–325.

Please see our response to point B. Our fibrils are not actin filaments, although we understand the reviewer's logic for why they could be.

d) There are some additional observations and questions pertaining to point 1.

The filaments seem to be polarized, like actin comets in the case of *Listeria*, perhaps indicating a preference of the NTD for the polymerizing or depolymerizing end of actin?

We cannot comment with any certainty on whether the fibrils we observe are polarized. Biophysical studies on the fibrils will be the subject of future study.

Why are the filaments not formed in the context of full-length coilin, i.e., in the R36A mutant, which also goes to the nucleus (compare Fig. 3B and 4B)?

We agree that the R36A mutant in the context of full length coilin does not form filaments when transfected into MEFs and presume that interactions of the C-terminus with other components, such as snRNPs, overrides this phenotype. This is why it is so important that in Fig 6, wild-type full-length coilin prominently joins the NTD fibrils! This shows that the NTD in full length coilin can interact with the test NTDs and participate in fibril-forming activity of NTD R36A. This is consistent with the possibility that the NTD in full-length protein can form fibrils but we do not currently have an assay that can directly assess whether fibrils are present within the observed puncta/CBs. We have now explicitly stated the rationale for this interpretation in the context of figure 6 in the Results and the Discussion.

Could the point mutations affect the structure of the ubiquitin fold itself rather than its interaction?

We think this is unlikely, for several reasons: First, the predicted structure places the affected amino acids on the outside surface of the structure. Second, if each amino acid change were destabilizing the ubiquitin fold, then likely that all of the amino acid changes would have similar phenotypes; instead, the R8A, R36A and D79A mutations have independent phenotypes, and the R8A/D79A double mutant has the opposite phenotype of R8A and D79A alone (Fig S7). Third, the fact that R8A and D79A mutant NTDs can interact with WT endogenous, full-length coilin indicates the NTD is still functional. We have added this reasoning to the second paragraph of the Discussion.

Even the nuclear puncta are sufficiently large to harbor polymers, not only oligomers.

We agree. Our hypothesis does not require that the puncta are formed of oligomers. We have now drawn a working model and figure 7, which should be seen as an hypothesis to be tested further rather than a proven fact. This working model does illustrate the data as we show them in this manuscript, enabling the reader to follow the logical arguments we present in the discussion.

2. Most of the experiments are performed in coilin $-/-$ MEFs known for their lack of normal CBs, but with “residual CBs”. In those cells, coilin is only deleted from the second exon onwards, such that the NTD alone could still be expressed. In fact, the authors from that study were so concerned with that possibility that they exogenously expressed a GFP-tagged NTD (GFP-mcoilin^{KO}), which localized to CBs in wild type MEFs but not in coilin $-/-$ MEFs. Interestingly, close inspection of its expression in the coilin knockout MEFs shows something like filaments in the cytoplasm (Fig. 7A of that study). Regardless, it is a serious problem if the NTD were already expressed in the cells used for most of the studies. This point needs to be clarified.

Like Matera and colleagues, who produced these cells, we have performed western blots, looking for the potentially expressed NTD in the mouse knock out cells. We have never observed a band of the expected molecular weight. However, like those previous authors, we cannot exclude the possibility that some molecules of the NTD are naturally expressed. Even if they were, the results that we obtained with tagged transfected coilin NTD still stand alone. This is because the mutations we have made give us information about the requirements for recruitment of Nopp140 and for the attainment of the punctate form within the nucleus (as opposed to fibrils).

3. It is questionable to study membrane less organelles (MLOs), which are extremely sensitive to variation in concentration of their components, by exogenous expression of some of the components. Obviously, the mere expression of coilin or NTD will affect CBs raising questions about the reported observations.

We agree. We have now included analyses of concentration dependence data (see Fig S1c and Fig S6d) and have elaborated a longer answer to this point in response to reviewer 2, who made more extensive comments on this issue. Please see the response to reviewer 2.

4. As interesting and unexpected the NTD filaments may be, they are not physiological but artificially induced. This does not mean that they are not a worthwhile phenomenon to study or learn from but lessens the general interest.

We agree that NTD fibrils are non-physiological, because the NTD has been expressed independent of the rest of the coilin molecule and localized to the cytoplasm. However, this does not exclude the possibility that fibrils form normally, especially since the wild-type NTD robustly forms fibrils. Indeed, our data indicate that endogenous coilin participate in fibril formation with wild-type and mutant NTDs (Fig 6). Our new discussion acknowledges that the role of coilin’s fibril-forming capacity in CB assembly requires further investigation.

5. Although the link between Nopp140 and coilin is interesting, there are several holes in that story.

a) NTD-NLS and Nopp140 colocalize exactly in nucleoli and CBs. This is not observed for the NTD alone, which presumably stays anchored in the cytoplasmic filaments despite being small enough to diffuse into the nucleus. Rather than the NTD interacting with Nopp140, it is equally possible that it is the NLS part that interacts with Nopp140. In fact, the very same SV40 NLS was used to identify and purify Nopp140 and shown to colocalize with Nopp140 upon incubation with permeabilized cells. If this is what is observed in this study, the interpretation of the results is wrong.

We agree that the use of the SV40 NLS could potentially confuse the activities detected in the fusion protein. However, the fact that the interaction with Nopp140 (by both FRET and pulldown measurements) is abolished by a single amino acid mutation in the NTD portion of both the fusion protein *and* the full length coilin construct lacking the SV40 NLS indicates that Nopp140 is interacting with the coilin NTD and not the SV40 nuclear localization signal.

b) The optodroplets of the Nopp140 intrinsically disordered region (IDR) induced by light in nucleoli and CBs are quite interesting (Fig. 6C). However, the interpretation is illogical. Before light induced dimerization, the Nopp140 IDR is diffusely localized throughout the nucleoplasm but after light activation, it matches the localization of endogenous Nopp140 in nucleoli and CBs. This suggests that dimerization of the Nopp140 IDR is required for its association with endogenous Nopp140, not with coilin, which is only in CBs.

In this experiment, we are comparing the potential of the coilin and Nopp140 IDRs to form biomolecular condensates *in vivo*. Because the Optodroplet assay is carried out in 3T3 cells, which lack Cajal bodies, it is not the case that Nopp140 IDR is associating with pre-existing Cajal bodies. It is possible that endogenous coilin is present in the induced clusters, just as coilin was present in clusters formed by the SMN tudor domain in our recent paper (Courchaine et al 2021). Regarding Nopp140, we do not agree with the interpretation that Nopp140-Cry2 interacts with endogenous Nopp140: A) it could already do that without the light stimulus (and it does not), and B) light activation only induces Cry2-Cry2 interactions. We believe the correct interpretation is that light induced dimerization leads to clustering of the Nopp140 IDR and not the coilin IDR. This enables us to predict that coilin NTD interaction with Nopp140 should lead to clustering, and this expectation met by many of the findings in this paper, including the dependency of Cajal bodies on Nopp140 expression (Figure 2). We have attempted to ensure that the writing of these sections of the results and discussion emphasize this logic.

c) Fig. 2 claims that Nopp140 is essential for proper CB assembly. All CB components are required for “proper” CB assembly. For example, it was published that knockdown of Nopp140 in stable cell lines causes a displacement of all scaRNPs from CBs leaving behind coilin, snRNPs, and SMN in condensates that are no longer “proper” CBs. Regardless, the knockdown of coilin and Nopp140 (Fig. 2) by transient transfection of siRNAs, where differential effects on coilin and SMN puncta are observed, needs to be better controlled. For example, in immunofluorescence panels E and F, it is not clear which cells are silenced, partially silenced, or not at all silenced. To

determine the effect on Nopp140, a Nopp140 stain is required. In panel F, the line profile of a cell that is not silenced at all seems to be shown claiming increased coilin in the nucleoplasm after Nopp140 targeting. Obviously, this needs to be controlled. In the quantification panels B and C, only positively identified silenced cells should be counted.

We thank the reviewer for bringing these points to the foreground. We agree that the claims of “proper” CB assembly or “proper” appearance of objects in the cell is an unspecific way of referring to an effect. We have dropped all reference of “proper” CBs. We have addressed the reviewer’s concern by automating the detection of CBs per nucleus, using coilin as a marker. This has allowed us to examine many nuclei objectively and unambiguously. When we plot the effect of coilin and Nopp140 depletion, we see that the number of CBs per nucleus drops from ~2 in the control to zero as a median value. This is a striking and statistically significant effect. Thus, we disagree with the suggestion that “coilin, snRNPs, and SMN [remain] in condensates”; please see new Fig S2a showing that snRNPs are dispersed when Nopp140 is depleted. The reason we undertook the experiment is that a recent paper from the Meier lab showed remaining CBs after a genetic knockout of coilin; because Nopp140 is an essential protein, we could not understand how these cells could actually be a knockout. Therefore, we can conclude that CBs require Nopp140 for their assembly and/or maintenance in our hands and in our cells. Given the statistical significance of this result from data collected on a large number of cells (see new text in the results section and Fig 2 legend), we feel it is not necessary to determine the level of knock down in each individual cell as the reviewer suggests. If the effect had been less pronounced, it may have been necessary to sort through the cells to find the knockdown cells. Finally, we have removed the line scans which we feel were previously confusing. The point of this figure is simply that Nopp140 depletion leads to the loss of CBs.

d) Nopp140 is only targeted by siRNAs in cells with endogenous coilin. Would CBs still form if coilin were transfected into coilin -/- MEFs after Nopp140 silencing?

We have not tried that experiment, because we have provided several lines of evidence that Nopp140 is required for condensate formation by NTD^{NLS} as well as full length coilin.

e) As already stated, NTD-NLS colocalizes with Nopp140, e.g., Fig. 1F. Then why is the NTD-NLS in Fig. 1C only in puncta and not in nucleoli? Also, Fig. 1F is shown twice, again in Fig. S1E upper panels. Are there not sufficient images to show different nuclei with the same result? Does the localization vary?

We apologize for any confusion concerning this point. There is indeed NTD-NLS protein signal in nucleoli, as the reviewer suggests, and we now address this explicitly in several experiments. This is visible in Fig 1c, new Fig S1d showing double staining with nucleophosmin, new Fig 3d, new Fig 5b (colocalization with nopp140). We apologize for the inadvertent use of the same image to illustrate the same experiment. We have substituted a second image now so that readers have two examples; we do have many images depicting localization. We have not observed variability in Nopp140 colocalization with NTD^{NLS} puncta.

6. There appear to be some inconsistencies with the FRET data. The FRET efficiency of full length coilin is about two-fold higher when the FRET labels are at its N versus C termini, and

close to zero when put on opposite ends. When the NTD is replaced by the GCN4 dimerization domain, the N terminal labels still yield FRET, but the C terminal ones result in negative FRET, clearly identifying a difference in behavior between the NTD and GCN4. Therefore, the data shows the opposite of “confirming the functionality of the GCN4 dimerization domain (Fig. S1b)” (page 6). Hence, and counter to what is claimed, it is not surprising that GCN4-coilin behaves differently from coilin with the NTD when transfected. Similar misleading setups are used on other occasions in the manuscript.

We apologize for what seems to be a misunderstanding regarding our method. We are performing indirect FRET, otherwise known as acceptor photobleaching (see Karapova et al., Stanek et al., 2004 and Dundr et al., 2004), which yields an apparent FRET value. In this method, performed on fixed cells, one images both fluorophores, then bleaches the acceptor fluorophore and then reimages both fluorophores. Apparent FRET is calculated by the fluorescence of the donor after bleaching relative to the fluorescence of the donor before bleaching. When substantial FRET occurs, the fluorescence of the donor will be much greater after bleaching. The reason it is possible to get a negative value is that bleaching of the donor fluorophore between the first and the last image (purely due to repeated imaging) will reduce the value below zero if there is no FRET. You can see this in the first three control measurements, where the negative FRET values show bleaching of the donor in the case where there is no FRET (negative controls); in the positive control (CFP-YFP fusion protein) FRET drives the value in the positive direction. While some investigators might normalize to the negative control value, we feel it is more straightforward to show the actual values. We have now added further explanations and references for this method and its analysis to the methods section.

FRET depends strongly on the orientation and the distance between the two fluorophores, so lack of FRET is uninterpretable (e.g. the second GCN4 construct). Positive FRET is meaningful. Regarding whether GCN4 replacement of the NTD induces FRET, the answer is YES, because Δ NTD yields -5% compared to +10% apparent FRET, so GCN4 mediates interaction. Note that 10% apparent FRET is a large effect. Regarding the difference between full-length coilin and the GCN4 construct, we agree GCN4 yields less FRET but we cannot say the reason why. It could be that the two different sets of interacting domains alter the orientation of the fluorophores. Note that the experiment shown in Figure 1b is with *no* fluorescent tag. Please note also that we now include relative concentration data in Fig S1c at the request of Reviewer 2, which reinforce the conclusions we have drawn.

Moreover, since FRET seems to be a bit capricious, it would be important to show permutations with N and C terminal labels for the Nopp140 coilin FRET, i.e., as for coilin in Fig. S1B. Obviously, the FRET results must be taken with a grain of salt.

We disagree on this point and believe this may be a misunderstanding due to our failure to explain those measurements in the supplementary figure legend. The N and C terminal labeling is used by us as a quality control to ensure our assay is detecting a range of FRET signals working in our hands as expected (see Stanek et al 2004). We have now modified the legend for clarity.

Taken together, our FRET experiments were performed with a high degree of rigor and replicated in a laboratory accustomed to performing these assays, which we have published previously. The FRET data show that coilin-coilin and coilin-Nopp140 pairs are within 10nm; in contrast our validation of these data by co-immunoprecipitation have no distance implications. The proximity required for FRET makes it more likely that these interactions are direct, though it does not prove the point and we do not interpret the data in this fashion. Instead, we have been able to test the effects of our single amino acid mutations on these interactions by FRET as well as pulldown, aiding our interpretations of the assembly phenotypes.

7. The dominant negative effects of the NTD constructs in HeLa cells with coilin and CBs are interesting (Fig. 5). Not described is the effect of the wild type NTD-NLS, which colocalizes with Nopp140 in nucleoli and CBs and which disperses endogenous coilin from CBs (Fig. 5D). The data are quite clear as three transfected cells (on the right) and three untransfected cells (on the left) are shown. Perhaps the transfected or the untransfected cells should be labeled in those panels. Anyway, does this mean that coilin is only held in CBs through its NTD? An interesting result that should be pointed out.

This is indeed the case. Hebert and Matera published that coilin no longer localizes to the CB when the NTD is depleted (Hebert, M. D. & Matera, A. G. Self-association of coilin reveals a common theme in nuclear body localization. *Mol Biol Cell* **11**, 4159-4171. (2000)). The suggested competition is consistent with this point. We have added a sentence to this effect in the results section.

In summary, the authors approach a complex problem and try to put an interesting spin on it. However, the manuscript must have gone through many iterations, as figures are mislabeled (by small letters in the text but capitals in the figures) or altogether not referred to (Fig. S3B).

We apologize for any inconsistencies in the use of small versus capital letters in figures. We have thoroughly check throughout the main and supplemental figures to ensure that no such errors remain.

Regarding S3B, what does it mean “soluble” versus “insoluble”, does that draw the immunoprecipitations in Fig. 3E into question?

The figure legend for this panel was previously inadequate. As referred to in the results, we are showing the starting material for the immunoprecipitation analysis carried out in Figure 3 (now Fig 4). Figure S5B shows the abundance of Nopp140, coilin, and the different constructs in the soluble fraction of the lysate versus the insoluble fraction. The reason this is important is that a band could be missing if it became insoluble by e.g. forming a large aggregate in the lysate. These controls show that when the amino acid mutations reduce Nopp140 pull-down, it is not because Nopp140 is missing (i.e. aggregated) in the lysate. We have amended the figure legend to be more self-explanatory.

Reviewing was not made easier by the lack of descriptions of material and methods, e.g., none of the antibodies are described nor is STET.

We have carefully reviewed the materials and methods, which now includes STED and a list of the antibodies used.

After a thorough overhaul including proper controls and experiments as pointed out above, I would certainly look forward seeing some of this perplexing data published, but not in the present form.

We thank the reviewer for his/her overall positive view of our work!

Reviewer #2 (Remarks to the Author):

In “Coilin oligomerization and remodeling by Nopp140 reveals a hybrid assembly mechanism for Cajal bodies” Courchaine, Machyna et al., focus on elucidating the molecular principles of Cajal body (CB) assembly that have largely remained elusive despite the long standing knowledge of the essentiality for Coilin. They provide some conclusive insights for the importance of both Coilin oligomerization and for the interaction between Coilin and Nopp140 in CB assembly within double knockout coilin MEF cells. Given the well known function of Cajal bodies and well documented importance they have in spliceosome biogenesis their mechanistic insights will be impactful. That said, I have significant concerns regarding the lack of description behind the microscopy data and the often poor quantification of it and the extent to which constructs are expressed. It is essential to understand the extent to which the authors express various constructs to be able to understand the extent to which a given construct can or cannot promote assembly of CBs, CB-like puncta, or droplets. Without changes in these two elements, as it stands their data as presented cannot fully support the major elements of their manuscript.

The reviewer indicates that his/her major constructive criticism concerns the quality of our descriptions of microscopy data and quantification of expression. We have answered the reviewer’s specific points below, by adding significant data relevant to both aspects.

Concerns:

1) The major overarching theme behind my primary concerns with the manuscript pertain to the quantification of the microscopy data. Throughout the text, statements such as X forms or does not form CB/fibers are frequently stated. Representative microscopy images are provided typically with a scale bar for intensities. However, little discussion is mentioned for if and if so how the absolute intensities can be related, even to images which are adjacent. Do the authors always use the same settings for the same microscopes? Alternatively, to increase dynamic range, do they have methods to reference the intensities to one setting (e.g. calibrated function/table to convert from any setting to a reference one)? Having this is important because- as it stands- the readers cannot assess if one construct expressed poorly, and thus precluded CB/fiber assembly or if 10X of expression was needed compared to another to result in the formation of CB/fibers (see concern 2 for more on this). As will be mentioned specifically in multiple locations below the authors need to address how the expression and localization seen via microscopy is quantified in cases where constructs are compared. Otherwise there remains significant doubt that the results aren’t dependent on other variables (e.g. microscopy settings or what tag was used).

For all images except those in Figure 3b, 4b, 5a, 6, S1f&g, S3, S5a, and S7. images were taken with Leica SP8 laser scanning confocal microscope. A white light laser was tuned to 594 nm at 22.1% (for coilin, myc imaging) and 488 nm at 10% (for Nopp140 imaging). For the images that compare FL coilin, coilin delta97, and coilin-GCN4 (Figure 1B, S1c) the PMT gain was set to 515.3 V and offset -0.4% for all images. For the images that compare NTD-NLS WT, R8A, R36A, D79A, and D40A (Figure 5B, S7c), the PMT gain was set to 636.2 V and offset -0.4%. In all these figures, a grayscale or color bar is provided to indicate the intensity value range of the image. If images have not been intensity-matched, multiple color bars are provided to indicate the range for each image. For figures 3b and S3, imaging was performed using the Bruker Opterra II Swept Field Instrument. For figures 4b, 5a, 6, S1f&g, S5a, and S7, imaging was performed on the DeltaVision. We have indicated all of this information in the methods section.

2) In the field of phase transitions (whether first order, e.g. LLPS, or higher order, e.g. micellization), the essential components of such processes have a concentration whereupon the phase transition occurs often referred to as the saturation concentration or C_{sat} (but also via other names such as the CMC for micelles). This has received significant attention in the field of membraneless organelles//biological condensates (A. Klosin et al., Science 2020; Riback et al., Nature 2020; J. M. Choi et al., PLOS Comput. Biol. 2019). Thus where possible the authors should attempt to determine if there exists a threshold where CB, CB-like, or fibers occur. For example, is Coilin-FL essential for Cajal bodies? If so below some nucleoplasmic C_{sat} there should be no cajal bodies. Similarly, when assessing constructs that do not support assembly (e.g. Coilin-deltaNTD) the max overexpression where this is true, relative to a construct that does support assembly such as FL-Coilin, should be reported. If for example Coilin-deltaNTD was expressed only 0.5, 2, or 100 of the nucleoplasmic levels that the FL coilin needed to assemble, this would be no, minimal, high, support for the stated claims that Coilin-deltaNTD cannot drive CB formation. As such, this will allow the readers to assess to what extent the conclusions that various constructs do not support assembly can be assessed. To aid the authors, I will try to mention all the locations where I think this is relevant below.

Typically, authors only calculate C_{sat} values for *in vitro* experiments with purified protein components in buffer. When performing *in vivo* studies of biomolecular condensates, one can only relatively estimate whether concentration correlates with condensation in cells. For example, in Guillen-Boixet, Cell 2020, the authors determine a C_{sat} for their *in vitro* experiments (Figure 2G) and provide a relative saturation concentration for their *in vivo* experiments (Figure 1C). We have calculated a relative saturation concentration to compare FL Coilin, Coilin delta97, and coilin-GCN4 constructs – in Fig S1c, and NTD-NLS WT, R8A, R36A, D79A, and D40A constructs – in Fig S6d. These curves demonstrate that the observed morphological effects are not merely a function of expression/cellular concentration, but rather a fundamental biochemical effect based on construct identity (based on the construct's ability to interact or not with interaction partners). To further address issues of expression differences between constructs being compared, we have performed western blots and calculated transfection efficiencies for the two groups of constructs listed above in figures S1a and S6b. Clearly, NTD-NLS R8A has very low transfection efficiency (Fig. S6a, b & e) and there for may be lowly expressed in the cells, which may explain the diffuse phenotype. However, NTD-NLS D79A has a similar phenotype to R8A and is more comparably transfected to the other constructs

tested, consistent with our interpretations. Overall, we feel these added data strengthen our findings by clarifying issues concerning protein concentration.

3) Keeping comments 1&2 in mind. In figure 1B, the amount of overexpression where FL-Coilin forms CBs should be reported and contrasted with the extent to which Coilin-deltaNTD and with the GCN2 dimer were overexpressed. How far did the authors push the GCN2 dimer construct to assess if it was sufficient at high levels to form CBs or CB-like structures.

We have repeated the transfection of these coilin constructs in the new Fig 1b, where the cells were imaged by confocal microscopy and displayed in grayscale for further clarity of the granularity or smoothness of the protein distributions. Figure S1a demonstrates that the expression of Coilin FL, Coilin-deltaNTD, and Coilin-GCN4 are relatively similar, and Fig S1b shows that the GCN4 construct FRETs within 2-fold of WT coilin, while Fig S1c makes clear that we have analyzed WT and mutant coilin over an ~30-fold range. Our relative saturation concentration estimation reinforces that the Coilin FL construct has a lower saturation concentration (Fig S1c, described more in depth in response to point 2).

4) Can the authors comment on the punctate pattern in the nucleoplasm for many coilin constructs? Is this an artefact of the antibody staining (vs. fluorescent tags). Also it's difficult in many cases to determine what is being imaged (which antibody/fluorescent tag etc). Reporting these in the figure or figure legend would be incredibly useful for the reader.

We believe this punctate pattern was an artefact from an older DeltaVision instrument or an older antibody. When these experiments were performed again more recently with a new anti-coilin antibody on the Leica SP8 instrument, this punctate pattern was not observed, even after deconvolution (e.g., see Fig 1b). The Cajal body phenotype observed is the same regardless of instrument or antibody used.

5) The authors seem confident that the fibers are simply driven by interactions between Coilin-NTD molecules. What is the support for this? One way I could envision demonstrating these are driven by the oligomerization of Coilin is by confirming that there is a fixed Csat for fibrillization. A lack of fixed Csat is indicative of other components driving/impacting the stability of assembly (for example Riback et al., Nature 2020; Posey et al., JBC 2018). Without evidence, the language should probably be toned down accordingly.

We have accordingly adjusted the language to reflect that this interaction may not be direct or may be influenced by other molecules.

6) Could the authors comment on the amount (e.g. Csat) of NTD-NLS needed to form CB-like structures vs. FL needed to form bonafide CBs. Do the other domains in Coilin contribute to stabilizing CBs (i.e. is the Csat nearly the same)?

This is an interesting and important question to answer, and we intend to answer this question *in vitro* with purified FL coilin and coilin NTD in the future. In the current *in cell* system, we do not feel comfortable comparing the relative saturation concentrations (as in Fig S1c or S6d) of FL and NTD because they are detected using different primary antibodies.

7) The authors make specific claims about more Coilin being in the nucleoplasm when referencing fig 2d-f. Addressing concern (1) above is fairly important to confirm that these can in fact be compared quantitatively as they intend.

In response to this comment, we have removed these claims. On close inspection, the data collected for this experiment is not sufficient to support the specific claim on nucleoplasmic levels between these samples due to the dynamic range measured for these samples. The important point of this figure is that CBs depend on nopp140, and so we have removed the line sub plots to focus the message of the figure.

8) With respect to concern (2) above, how is the total overexpression compared between the ALA mutants in figure 3 and corresponding supplement figures. For the ones which do not support CB formation at any concentration what is the max overexpression? Can the authors rule out that some variants just do not simply express weaker than others? Ideally if the authors could compare the Csat for the ALA mutants with WT this would better quantify mutants which may destabilize CB assembly but do not fully eliminate it (e.g. because their mutations may only weakly perturb the unknown interaction surfaces).

This figure and supplemental figure (now Figs 4 and S5) show images that are from an alanine mutation screen. All cells were transfected with the same amount of DNA, however this does not alone determine protein concentration in a nucleus (transfection efficiency, protein degradation could also play a role). We acknowledge that there likely is variability in the protein concentrations between constructs. The intention of the screen was to identify interesting mutants to follow up on using interaction assays and microscopy. For the mutants we followed up on (R8A, R36A, D79A, D40A), we have included metrics to better understand the relative protein levels in each cell (see Figure S6d, relative c-sat estimation).

9) Figure 4 and 5 needs to be addressed with respect to concerns 1&2 above in terms of the extent to which the constructs are expressed.

We have addressed this above.

10) The authors assert that R8A in figure 4B is not forming CB but going to nucleoli. Can the authors elaborate how they assess this? These puncta look similar to CBs to this reviewer.

We have added a panel (see Fig 5b) that includes co-staining of NTD-NLS constructs with Nopp140 to address this concern.

11) Can the authors clarify or comment on the result as to why the NTD-R8A which fails to oligomerize and interact with Nopp140 would impact Cajal bodies in Fig 5? Does this suggest another interaction partner?

This is an interesting point. We address this by doing a double mutant (R8A and D79A) experiment, where we see that when you abolish coilin interaction on both “faces” of the molecule, endogenous Cajal bodies are unaffected (Fig S7). We feel the most parsimonious

interpretation is that mutant NTD fails to sequester endogenous coilin molecules, because they can't interact at all. We tried to more clearly explain this in the new version of the paper.

12) Is the disassembly of Cajal bodies due to truncated Coilin dose-dependent similar to the competition assay of Figure 3 in Sanders et al., Cell 2020.

If we understand correctly the reviewer is asking whether the expression of non-functional coilin is able to “cap” the interaction network of these proteins. We believe the most applicable data is found in Fig 6, where NTD fragments have dominant negative effects on endogenous CBs. Figure S7B suggests that multiple mutations, R8A and D79A together, result in rescue of CBs and thus the NTD itself is the meaningful part of the network, and single mutations act as “end capping.” The truncated deltaNTD coilin does not have this effect on Cajal Bodies in our hands, and this is what was found in Hebert and Matera (2000). As to whether these effects are dose dependent, we have not measured this but our quantification in Fig 6 suggests that the effects of R8A and D79A are highly penetrant.

13) It has been observed that overexpression of Coilin results in larger Cajal bodies in HeLa cells and is stabilized by heterotypic interactions between coilin and other components (Riback et al., Nature 2020). While the number does not seem to change as reported in Figure 5 with the overexpression of NTD-NLS. Does the overexpression of Coilin FL and/or NTD-NLS result in increases in size to CBs in these cells?

We are overexpressing Coilin FL and NTD-NLS in these cells, and we do not see abnormally large Cajal bodies. Perhaps studies showing changes in size are overexpressing to a higher level.

14) Given the concentration dependence of the Cry2 system for light triggered activation as reported in Shin et al., Cell 2017, it seems important to address the relative overexpression levels (e.g. concerns 1&2 applied to the Cry2 data). Also was Cry2-CoilinDeltaNTD also tested? It seems that this would better mimic the architecture than CoilinDeltaNTD-Cry2. Given the claim that the homotypic interactions between the Nopp140 IDR may be required for CB assembly (due to the results of the Cry2 assay with the Nopp140 IDR) and not simply that Nopp140 yields additional multivalency and prevents fiber-like oligomers, it would seem the correct orientation would better mimic the FL protein.

We found the coilin Cry2 constructs to have very low transduction efficiency and thus our observations are limited to cells in the expression range shown in Fig 3b and c and Fig S3. We believe that this concern is more directly addressed by the GCN4 experiment in Figure 1. While we agree an N-terminal architecture would be a better mimic, we believe the assay as designed adequately shows that coilin without its NTD does not act as a homotypic condenser in the concentration range tested.

15) The authors imply that the occurrence of coiled electron dense fibers within CBs by EM as indicative of a lack of liquid nature of CBs. Given that fibers can undergo LLPS into liquids (e.g. Weirich et al., PNAS 2017), presenting the EM as in disagreement with CB as being liquids as

opposed to reporting the internal structure of CBs seems inaccurate and misleading.

Thank you for this comment. We have now included this possibility in the discussion.

16) The authors state that 50% of Coilin molecules are immobile and cite reference 11 (Dundr et al., JCB 2004). This is incorrect as reference 11 as done could not assess immobility; the authors of 11 of that work make no such conclusions. As such these authors need to correct this (or correct this reviewer on why they make such claims when the ref 11 authors do not). In fact others (Bártová et al 2014 - <https://doi.org/10.4161/nucl.29229>), report significant recovery after <1min. Furthermore, I don't quite understand why the authors' model requires Coilin oligomers to be immobile. Small oligomers such as NPM1 show >90% recovery on the min timescale (Zhu et al., PNAS 2019). If this is essential, the authors could FRAP CBs and NTD coilin cytoplasmic fibers to validate that Coilin oligomers aren't in fact dynamic in these contexts (being sure to correct for the fraction of molecules bleached as bleaching a whole CB often bleaches a significant amount of the total fluorescent Coilin- a problem I think ref 11 was trying to avoid).

We agree and have removed this sentence from the discussion. The models we consider and discuss do not require multimers to be immobile.

17) Minor- Supp figure 4D - Is Sat3-RFP a mistake?

Yes, thank you - this has been corrected.

REVIEWER COMMENTS

Reviewer #1 (Remarks to the Author):

This is a review of a revised manuscript originally submitted nearly a year ago. So much has changed and been rearranged in this version, including addition of a new co-first author, that this is a new review. Where appropriate, it will also address the responses to the previous review.

The main conclusions of the current manuscript are that the N-terminal domain (NTD) of the Cajal body (CB) marker protein coilin can form artificial fibrils in the cytoplasm. However, when targeted to the nucleus by an exogenous nuclear localization signal (NLS), NTD-NLS accumulates in puncta together with the nucleolar phosphoprotein Nopp140. The intrinsically disordered repeat domain of Nopp140 appears to form condensates when forced together. The final model suggests that NTD-NTD mediated assemblies make multivalent contact with Nopp140 forming biomolecular condensates.

The revised manuscript generally tried to address the points raised and reads much better and more linearly. Nevertheless, my enthusiasm is tempered by the fact that the two main observations, the self-interaction of the coilin N-terminus and its interaction with Nopp140, have already been published over 20 years ago, even if in different contexts and without description of the cytosolic NTD fibrils. Those two papers are:

Hebert, M. D. & Matera, A. G. Self-association of Coilin Reveals a Common Theme in Nuclear Body Localization. *Mol Biol Cell* 11, 4159–4171 (2000).

Isaac, C., Yang, Y. & Meier, U. T. Nopp140 Functions as a Molecular Link Between the Nucleolus and the Coiled Bodies. *J Cell Biology* 142, 319–329 (1998).

Please, see some of my specific comments below, more or less in order of appearance (all references to figures refer to the revised ms).

1. Page 7, final paragraph: NTD-NLS does form puncta in the nucleoplasm of coilin $-/-$ -MEFs that are devoid of SART3, SMN, SmB", but not, unlike what is claimed, devoid of fibrillarin. Even if the fibrillarin stain is weak, there is a clear signal of fibrillarin in residual CBs of the $-/-$ -MEFs (Fig. S1g, compare the inserts). This has been well-documented in the manuscript that first described those cells. Residual CBs contain Nopp140 and fibrillarin. In fact, Nopp140 is always found in association with sno/scaRNPs (for which fibrillarin is a marker), even in artificially induced intranuclear R-Rings that include coilin [Isaac, C., Pollard, J. W. & Meier, U. T. Intranuclear endoplasmic reticulum induced by Nopp140 mimics the nucleolar channel system of human endometrium. *Journal of Cell Science* 114, 4253–4264 (2001)]. Thus, transfected NTD-NLS could bind to Nopp140 in residual CBs.

2. P. 8, first paragraph: Apparently, based on a misunderstanding/misinterpretation of published data, the recently published Nopp140 knockdown cells are implied by "CRISPR targeting of both alleles of Nopp140" to be Nopp140 knockout cells. As clearly stated in those two papers [Bizarro, J. et al. Nopp140-chaperoned 2'-O-methylation of small nuclear RNAs in Cajal bodies ensures splicing fidelity. *Gene Dev* 35, 1123–1141 (2021); and Bizarro, J., Bhardwaj, A., Smith, S. & Meier, U. T. Nopp140-mediated concentration of telomerase in Cajal bodies regulates telomere length. *Mol Biol Cell* 30, 3136–3150 (2019)], and despite using CRISPR/Cas9 technology, only a Nopp140 knockdown but not knockout was achieved in those polyploid HeLa cells. As cited in those papers, a similar observation with Nopp140 targeting in HeLa cells had previously been ascribed to polyploidy. – BTW, both those papers should be cited, as the more recent one documents an actual function of CBs, the modification of snRNAs.

3. Response to criticism 5c: The same confusion is also obvious from the authors' response to point 5c "... because Nopp140 is an essential protein, we could not understand how these cells could actually be a knockout." Again, in those papers, it clearly states and documents that those are only Nopp140 knockdown and not knockout cells.

4. The Nopp140 knockdown experiments in the revised manuscript are still poorly documented. If siRNA is used for Nopp140 knockdown, then it needs to be documented by Nopp140 staining because siRNA transfection is notoriously heterogenous. For example, in Fig. 2d, in 8 siNolc1 cells, SMN is visible in only 2 and coilin in all. Thus, the statement that "SMN appeared relatively unaffected" is false. Similarly, "CBs were broadly reduced after Nopp140 depletion" does not apply. Importantly, because of a lack of Nopp140 stain, it is not known if and to what degree Nopp140 is knocked down in each cell. In Fig. S2a, where Nopp140 staining was performed and went from gray to less gray (those seem to be poor Nopp140 antibodies), SMN is lost from 3 out of 5 cells. And in S2b, coilin goes from being present in only 4 out of 6 cells, to not being present in 6 cells (see also the points on quantification and cell clustering). The fact remains that upon stable Nopp140 knockdown in the two papers cited above, CBs can still be detected by transmission electron microscopy, only the size of the individual granules in CBs shrinks by about half, which can be rescued by Nopp140 re-expression. Thus, Nopp140 knockdown does not lead to general loss of CBs.

5. The cells shown in Fig. 2 and S2 are growing in clusters/islets instead of uniform monolayers. It is well documented that CB number and appearance are influenced by growth conditions such as confluency and varies upon transformation status of cells and cell cycle phase [Spector, D. L., Lark, G. & Huang, S. Differences in snRNP localization between transformed and nontransformed cells. *Mol Biol Cell* 3, 555–569 (1992)]. This needs to be taken into consideration when culturing and staining cells, which brings me to quantification.

6. Based on this and the other reviewer's comments, the authors make a valiant effort to automate quantification of expression of their coilin constructs and of CBs in general. This is a difficult task because CBs differ in size and intensity within the same nucleus and between nuclei. Therefore, even simple masking of CBs is not a trivial task and tends to be inaccurate as some CBs are missed or thresholded out. More controls for that approach are needed. Similarly, the value of the novel plots of percentage of nuclear area over mean nuclear intensity of various coilin constructs is unclear. For example, if a nucleus contained a single CB that is very bright, then at a low percentage of nuclear area the mean intensity should still be high, which is not seen in Fig. S1c. In Fig. S6d, the NTD-NLS seems to occupy over 50% of the nuclear area in some cells, which seems massive and not physiological. In the adjoining figure S6e, the resolution is too poor to see anything. Finally, even if arbitrary, why does the mean nuclear intensity between figures S1c and S6d differ by some 1000-fold? This quantification is certainly not intuitive.

7. I must correct a false statement in the discussion on p. 16, "... the naming of 'coiled bodies' before their identity with Cajal bodies was known". Coiled bodies were renamed Cajal bodies by Joe Gall in 1999 [Gall, J. G., Bellini, M., Wu, Z. & Murphy, C. Assembly of the Nuclear Transcription and Processing Machinery: Cajal Bodies (Coiled Bodies) and Transcriptosomes. *Mol Biol Cell* 10, 4385–4402 (1999)]. Hence, coiled bodies were always known to be identical with Cajal bodies.

8. Similarly, on page 17 "...Nopp140 solubility during mitosis, which is caused by extensive phosphorylation by cdc2 kinase...". Although Nopp140 is phosphorylated by cdc2 kinase, its only "extensive" phosphorylation is mediated by casein kinase 2 at some 80 serines.

I could go on but end here. The authors only partially addressed my criticism to satisfaction. The paper makes some interesting points about the N-terminus of coilin including the cytoplasmic fibrils, but what are they, do they contain other proteins, which, can they be formed in vitro, and most importantly, do they play any role in the cell? Also, the point mutations identified in the NTD further aid in dissecting its (self)interactions. In the end, it is not clear what we learn from this study, do the NTD interactions contribute to CB formation – maybe? The working model in Fig. 7 seems to be an oversimplification as it is not clear how the fibrils would contribute to CB formation because they are cytoplasmic and would be too large to fit into a CB.

Tom Meier

Reviewer #2 (Remarks to the Author):

In my initial review, I pointed out concerns regarding (1) the lack of description behind the microscopy data and (2) a general lack of quantification regarding the extent to which the constructs were expressed (given that biomolecular condensates typically form via phase separation). The authors have corrected both points sufficiently and addressed each of my numerated concerns appropriately.

Response to Reviewers:

We thank both reviewers for their efforts to read and critique our manuscript anew. **Reviewer #2** said “In my initial review, I pointed out concerns regarding (1) the lack of description behind the microscopy data and (2) a general lack of quantification regarding the extent to which the constructs were expressed (given that biomolecular condensates typically form via phase separation). The authors have corrected both points sufficiently and addressed each of my enumerated concerns appropriately.” We thank the reviewer for this comment. and

We focus below on the comments of **Reviewer #1** (reviewer comments in gray, our response in black):

This is a review of a revised manuscript originally submitted nearly a year ago. So much has changed and been rearranged in this version, including addition of a new co-first author, that this is a new review. Where appropriate, it will also address the responses to the previous review.

The main conclusions of the current manuscript are that the N-terminal domain (NTD) of the Cajal body (CB) marker protein coilin can form artificial fibrils in the cytoplasm. However, when targeted to the nucleus by an exogenous nuclear localization signal (NLS), NTD-NLS accumulates in puncta together with the nucleolar phosphoprotein Nopp140. The intrinsically disordered repeat domain of Nopp140 appears to form condensates when forced together. The final model suggests that NTD-NTD mediated assemblies make multivalent contact with Nopp140 forming biomolecular condensates.

The revised manuscript generally tried to address the points raised and reads much better and more linearly. Nevertheless, my enthusiasm is tempered by the fact that the two main observations, the self-interaction of the coilin N-terminus and its interaction with Nopp140, have already been published over 20 years ago, even if in different contexts and without description of the cytosolic NTD fibrils. Those two papers are:

Hebert, M. D. & Matera, A. G. Self-association of Coilin Reveals a Common Theme in Nuclear Body Localization. *Mol Biol Cell* 11, 4159–4171 (2000).

Isaac, C., Yang, Y. & Meier, U. T. Nopp140 Functions as a Molecular Link Between the Nucleolus and the Coiled Bodies. *J Cell Biology* 142, 319–329 (1998).

We cited both prior studies in the first and second versions of the manuscript. We explicitly state that we are building off their findings in the present manuscript, which identifies which single amino acids are required for the interactions, develops tools for molecular studies based on those identifications, and discovers potential alternative structures that these proteins can occupy and form. Our study employs modern methods and detailed molecular analyses to deepen our understanding of this system. The reviewer states above and in the previous round of review our main findings are that the coilin N-terminal domain has previously undescribed assembly properties (fibrils etc), that these are subject to point mutations, and that these mutations have direct consequences for the essential CB protein Nopp140. Thus, we do not find any disagreement with the reviewer and do not understand what the second comment (that those findings have been known for 20 years) is asking of us.

Please, see some of my specific comments below, more or less in order of appearance (all references to figures refer to the revised ms).

1. Page 7, final paragraph: NTD-NLS does form puncta in the nucleoplasm of coilin -/ -MEFs that are devoid of SART3, SMN, SmB⁷, but not, unlike what is claimed, devoid of fibrillarin. Even if the fibrillarin stain is weak, there is a clear signal of fibrillarin in residual CBs of the -/ -MEFs (Fig. S1g, compare the inserts). This has been well-documented in the manuscript that first described those cells. Residual CBs contain Nopp140 and fibrillarin. In fact, Nopp140 is always found in association with sno/scaRNPs (for which fibrillarin is a marker), even in artificially induced intranuclear R-Rings that include coilin [Isaac, C., Pollard, J. W. & Meier, U. T. Intranuclear endoplasmic reticulum induced by Nopp140 mimics the nucleolar channel system of human endometrium. *Journal of Cell Science* 114, 4253–4264 (2001)]. Thus, transfected NTD-NLS could bind to Nopp140 in residual CBs.

We agree with the reviewer that residual CBs documented in Tucker et al., 2001 (<https://doi.org/10.1083/jcb.200104083>) prominently contain fibrillarin. Note that the cited reference above (Isaac et al) comments “data-not-shown” for this finding. In contrast, the fibrillarin staining in our cells is broadly nucleoplasmic and not well-localized to the NTD-NLS puncta compared to nearby fluorescence that is brighter and non-overlapping. Thus, we were not convinced of the presence of fibrillarin in NTD-NLS nuclear puncta from the staining in our own hands. Our NTD-NLS construct could bind to or generate puncta that accumulate some amount of fibrillarin; yet, the presence or absence of fibrillarin, a molecular component of snoRNPs, has little bearing on the rest of our conclusions. We did not attempt staining with every one of the more than 100 protein components of CBs, gems, residual bodies, etc. The reason we stained the puncta for Nopp140 is because of the Isaacs 1998 paper, showing that Nopp140 binds the coilin NTD, which we reference. Taken together, Reviewer 1 makes a good point that a reader could wonder if we think these are residual bodies. Therefore, we have completely re-written the paragraph at the bottom of page 7, addressing the fibrillarin staining and making further comments about expectations. We have also added information about scaRNAs and scaRNPs in the first two paragraphs in the introduction. We hope the reviewer will feel the additional edits we have made clarify these issues for all readers.

2. P. 8, first paragraph: Apparently, based on a misunderstanding/misinterpretation of published data, the recently published Nopp140 knockdown cells are implied by “CRISPR targeting of both alleles of Nopp140” to be Nopp140 knockout cells. As clearly stated in those two papers [Bizarro, J. et al. Nopp140-chaperoned 2'-O-methylation of small nuclear RNAs in Cajal bodies ensures splicing fidelity. *Gene Dev* 35, 1123–1141 (2021); and Bizarro, J., Bhardwaj, A., Smith, S. & Meier, U. T. Nopp140-mediated concentration of telomerase in Cajal bodies regulates telomere length. *Mol Biol Cell* 30, 3136–3150 (2019)], and despite using CRISPR/Cas9 technology, only a Nopp140 knockdown but not knockout was achieved in those polyploid HeLa cells. As cited in those papers, a similar observation with Nopp140 targeting in HeLa cells had previously been ascribed to polyploidy. – BTW, both those papers should be cited, as the more recent one documents an actual function of CBs, the modification of snRNAs.

First, we apologize for misrepresenting the situation of the CRISPR/Cas9 mutated cells from the Meier lab; this is an unusual use of genome editing that resulted in a knockdown, instead of a knockout. We have amended the text on top p8 to precisely describe the degree of knockdown and how they were made: “Partial knockdown of Nopp140 by stable CRISPR/Cas9 editing did not disturb CB formation^{6,41}, while two other studies showed that loss of Nopp140 is correlated with CB disassembly^{40,42}.” Second, our statement that two other studies reported a dependency of CB integrity on Nopp140 justifies the need for us to perform the experiment in our own hands and in

our cells. Third, we have now added the reference to Bizarro 2021 and apologize for its omission in the revised version. The first submission of our paper was 1 month prior to the publication of Bizarro 2021, yet long enough ago that we assumed it was cited (Bizarro et al. 2019 was already previously cited). Based on this comment, we have now also added more primary literature from both the Meier and the Kiss labs (particularly Jady et al 2003) to better describe the involvement of the CB in snRNA modification by scaRNAs, which we agree is a valid point.

3. Response to criticism 5c: The same confusion is also obvious from the authors' response to point 5c "... because Nopp140 is an essential protein, we could not understand how these cells could actually be a knockout." Again, in those papers, it clearly states and documents that those are only Nopp140 knockdown and not knockout cells.

Please see our response to point 2.

4. The Nopp140 knockdown experiments in the revised manuscript are still poorly documented. If siRNA is used for Nopp140 knockdown, then it needs to be documented by Nopp140 staining because siRNA transfection is notoriously heterogenous. For example, in Fig. 2d, in 8 siNolc1 cells, SMN is visible in only 2 and coilin in all. Thus, the statement that "SMN appeared relatively unaffected" is false. Similarly, "CBs were broadly reduced after Nopp140 depletion" does not apply. Importantly, because of a lack of Nopp140 stain, it is not known if and to what degree Nopp140 is knocked down in each cell. In Fig. S2a, where Nopp140 staining was performed and went from gray to less gray (those seem to be poor Nopp140 antibodies), SMN is lost from 3 out of 5 cells. And in S2b, coilin goes from being present in only 4 out of 6 cells, to not being present in 6 cells (see also the points on quantification and cell clustering). The fact remains that upon stable Nopp140 knockdown in the two papers cited above, CBs can still be detected by transmission electron microscopy, only the size of the individual granules in CBs shrinks by about half, which can be rescued by Nopp140 re-expression. Thus, Nopp140 knockdown does not lead to general loss of CBs.

With all respect, we wish to present a different point of view on these issues.

First, our knockdown levels are substantial. In our knockdown, we do not detect *any* background Nopp140 by western blot compared to the low levels in the CRISPR Knockdowns used by Bizarro *et al.* 2019 and 2021. This effect can be visualized in Figure S2a, where we see that knockdown is not absolute in a small population of cells but is reduced to minimal levels in many. We fail to see an argument for how our knockdown is insufficient to proceed with this analysis.

Second, representative images can only go so far to describe data, which is precisely why we provide both manual and automated quantification of CBs. Our results show a reduction in CBs by both methods, resting on the findings that A) coilin (one marker of CBs) no longer clusters to into discrete large puncta, as it does in WT cells (Fig 2). B) the remaining smaller coilin puncta are not proximal to SMN and they lack trimethyl guanosine (Fig. S2), two other well-established markers of functional CBs. Although we cannot guess how this would look by EM, reduced size and altered composition –embodied by these two features that we analyzed with care – constitute a strong rationale for concluding that CBs undergo disassembly in the absence of Nopp140.

As a final point on this, we think semantics plays a role in this discussion. We agree that claiming complete "loss" of CBs is a difficult point, and we do not do so. We ask "whether the coilin NTD–Nopp140 interactions contribute to CB assembly", we use the word "reduce" to characterize the effect on CBs that we measured, and we conservatively conclude that "Nopp140 is necessary for

the assembly and/or maintenance of CBs containing coilin and snRNPs". We hope the reviewer can agree that we are rigorously analyzing and describing our data, and that the data justify these conclusions. We comment further on quantification below under point 6.

5. The cells shown in Fig. 2 and S2 are growing in clusters/islets instead of uniform monolayers. It is well documented that CB number and appearance are influenced by growth conditions such as confluency and varies upon transformation status of cells and cell cycle phase [Spector, D. L., Lark, G. & Huang, S. Differences in snRNP localization between transformed and nontransformed cells. *Mol Biol Cell* 3, 555–569 (1992)]. This needs to be taken into consideration when culturing and staining cells, which brings me to quantification.

We have re-read the paper noted in this comment. Comparisons made were largely between cell lines derived from different tissues and with different transformation statuses, rather than confluency or cell cycle. Our study never makes quantitative comparisons between different cell lines. All of our comparisons are between the cells grown under the same conditions and over 100 cells are quantified per condition, as is standard in the field. We think it is unlikely that variables noted by the reviewer would impact multiple replicates and multiple fields of view.

6. Based on this and the other reviewer's comments, the authors make a valiant effort to automate quantification of expression of their coilin constructs and of CBs in general. This is a difficult task because CBs differ in size and intensity within the same nucleus and between nuclei. Therefore, even simple masking of CBs is not a trivial task and tends to be inaccurate as some CBs are missed or thresholded out. More controls for that approach are needed. Similarly, the value of the novel plots of percentage of nuclear area over mean nuclear intensity of various coilin constructs is unclear. For example, if a nucleus contained a single CB that is very bright, then at a low percentage of nuclear area the mean intensity should still be high, which is not seen in Fig. S1c. In Fig. S6d, the NTD-NLS seems to occupy over 50% of the nuclear area in some cells, which seems massive and not physiological. In the adjoining figure S6e, the resolution is too poor to see anything. Finally, even if arbitrary, why does the mean nuclear intensity between figures S1c and S6d differ by some 1000-fold? This quantification is certainly not intuitive.

We agree with the reviewer that image quantification is challenging and note that the other reviewer was satisfied by the steps we took. While any analysis pipeline is imperfect, we apply the same algorithm to all our samples and find clear differences. Furthermore, we also quantify our data by eye, and the results agree. This enables us to interpret our experiment.

The plots we present in S1c and S6d are not a novel quantification scheme. For an example, see Figures 2J&H in G.A. Corbet et al., 2021 (<https://doi.org/10.1242/jcs.258783>). If we are to assume that Cajal bodies obey the properties of most condensates observed in cells, the density of coilin in the condensed phase will remain constant as more is driven across the phase boundary. Thus, more volume will be taken up by the condensate.

We agree that edge cases in this analysis may not correspond to a physiological situation, but as with the case in *any overexpression experiment*, we are left to interpret physiology from trends and not edge cases. Our expression range and thresholded analysis show a clear and obvious trend that is consistent with our model.

We understand the comment about image quality, but the purpose of Figure S6 is to analyze transfection efficiency, not sophisticated morphology. The cells in figure S6e are the same condition as shown in the main text in Figure 5. The difference in range from figure S1c and S6d

is due to one dataset having been collected in 8-bit recording while the other data set necessitated 16-bit dynamic range. As noted by the reviewer, these values are arbitrary. We hope this clarifies the purpose of Fig S6.

7. I must correct a false statement in the discussion on p. 16, "... the naming of 'coiled bodies' before their identity with Cajal bodies was known". Coiled bodies were renamed Cajal bodies by Joe Gall in 1999 [Gall, J. G., Bellini, M., Wu, Z. & Murphy, C. Assembly of the Nuclear Transcription and Processing Machinery: Cajal Bodies (Coiled Bodies) and Transcriptosomes. *Mol Biol Cell* 10, 4385–4402 (1999)]. Hence, coiled bodies were always known to be identical with Cajal bodies.

We thank the reviewer for catching a badly worded sentence. This sentence has been revised to read "Interestingly, electron microscopy (EM) of CBs in somatic cells shows what appear to be coiled electron dense fibers that led to the naming of "coiled bodies" before they were rechristened "Cajal bodies" in honor of their first observer^{1,60,61}. Immunogold localization of coilin seems to decorate these coiled fibers when viewed by EM²⁶." We have also added the original reference to the 1999 paper in addition to Gall's classic review coining the name, which was previously referenced.

8. Similarly, on page 17 "...Nopp140 solubility during mitosis, which is caused by extensive phosphorylation by cdc2 kinase...". Although Nopp140 is phosphorylated by cdc2 kinase, its only "extensive" phosphorylation is mediated by casein kinase 2 at some 80 serines.

We have revised the text to reflect the appropriate kinase activity. This sentence now reads: "Nopp140 solubility during mitosis, which is caused by phosphorylation of ~80 serines by casein kinase 2, may play a role in CB dynamics^{41,64,65}."

I could go on but end here. The authors only partially addressed my criticism to satisfaction. The paper makes some interesting points about the N-terminus of coilin including the cytoplasmic fibrils, but what are they, do they contain other proteins, which, can they be formed in vitro, and most importantly, do they play any role in the cell? Also, the point mutations identified in the NTD further aid in dissecting its (self)interactions. In the end, it is not clear what we learn from this study, do the NTD interactions contribute to CB formation – maybe? The working model in Fig. 7 seems to be an oversimplification as it is not clear how the fibrils would contribute to CB formation because they are cytoplasmic and would be too large to fit into a CB.

Tom Meier

We appreciate that Reviewer 1 is satisfied by some of our accomplishments during the first revision, apparently including the experimental work that he asked for and that we provided. We feel that the reviewer's overall opinion of our manuscript is driven by concerns pertaining to the experiment on Nopp140 outlined in Figure 2. Likely due to differences in how the experiments were done, including possible differences between the cell lines, different interpretations were reached. Our methods lead us to conclude that acute depletion of Nopp140 decreases the number of CBs with the accepted properties in terms of size, number and composition. We go beyond Figure 2 to describe the basis of this interaction with coilin in unprecedented molecular detail.

We therefore wish to review the impact of the presented work. Since the Meier lab's paper in 1998, focusing on the role of Nopp140, and the Matera lab's paper in 2000, showing the NTD is

required for making or joining CBs, new information on the molecular function of the coilin NTD has been lacking. These are indeed classics in the history of Cajal body investigation, but – like all good papers – they raise many questions about the intermolecular interactions required to form a CB, and these have remained unanswered.

The findings we present here contribute new knowledge and conceptual advances, by demonstrating that the NTD is the necessary and sufficient part of coilin for nuclear condensation. We show that the NTD cannot act alone and provide evidence that its partner Nopp140 is necessary for condensation as well as Cajal body assembly and/or maintenance. We use point mutations to establish causal links between (i) the coilin NTD's self-association and nuclear condensation, and (ii) the coilin–Nopp140 interaction and nuclear condensation. Finally, we demonstrate that reducing NTD self-association or NTD-Nopp140 interaction disrupts endogenous Cajal bodies through the dominant negative effects of the respective point mutations, demonstrating that these molecular interactions can engage wild-type coilin. This answers the reviewer's question "do they play any role in the cell?". In this context, the working model presented in Figure 7 assists the reader in understanding how we currently envision the intermolecular interactions described. Twenty years ago, coilin was broadly referred to as a low complexity, intrinsically disordered protein. The work of others has established the molecular structure of the C-terminus and our work is consistent with a structured and bi-functional NTD, broadly changing our current view of how coilin forms CBs and how protein scaffolds might form biomolecular condensates in general. These advance are of broad interest to the field. The remaining questions, such as whether the fibrils "would be too large to fit into a CB", are worthy of future investigation but beyond the scope of the current study.

REVIEWERS' COMMENTS

Reviewer #1 (Remarks to the Author):

The authors have now adequately addressed my major concerns. Obviously, some questions remain, but this is all part of a healthy scientific discourse. I commend the authors for indulging me and significantly improving the clarity and scientific impact of their study.